# Improving Sparse Decomposition of Language Model Activations with Gated Sparse Autoencoders

**Senthooran Rajamanoharan**[*]
Google DeepMind

**Arthur Conmy**[*]
Google DeepMind

**Lewis Smith**
Google DeepMind

**Tom Lieberum**[†]
Google DeepMind

**Vikrant Varma**[†]
Google DeepMind

**János Kramár**
Google DeepMind

**Rohin Shah**
Google DeepMind

**Neel Nanda**
Google DeepMind

## Abstract

Recent work has found that sparse autoencoders (SAEs) are an effective technique for unsupervised discovery of interpretable features in language models' (LMs) activations, by finding sparse, linear reconstructions of those activations. We introduce the Gated Sparse Autoencoder (Gated SAE), which achieves a Pareto improvement over training with prevailing methods. In SAEs, the L1 penalty used to encourage sparsity introduces many undesirable biases, such as *shrinkage* – systematic underestimation of feature activations. The key insight of Gated SAEs is to separate the functionality of (a) determining which directions to use and (b) estimating the magnitudes of those directions: this enables us to apply the L1 penalty only to the former, limiting the scope of undesirable side effects. Through training SAEs on LMs of up to 7B parameters we find that, in typical hyper-parameter ranges, Gated SAEs solve shrinkage, are similarly interpretable, and require half as many firing features to achieve comparable reconstruction fidelity.

## 1   Introduction

Mechanistic interpretability aims to explain how neural networks produce outputs in terms of the learned algorithms executed during a forward pass [33, 34]. Much work makes use of the fact that many concept representations appear to be linear [14, 19, 34, 39], i.e. that they correspond to interpretable directions in activation space. However, finding the set of all interpretable directions is a highly non-trivial problem. Classic approaches, like interpreting neurons (i.e. directions in the standard basis) are insufficient, as many are polysemantic and tend to activate for a range of different seemingly unrelated concepts [7, 15, 16]. Within the field, there has recently been much interest [8, 11, 21, 22, 4] in using sparse autoencoders (SAEs; [32]) as an unsupervised method for finding causally relevant, and ideally interpretable, directions in a language model's activations.

Although SAEs show promise in this regard [26, 31], the L1 penalty used in the prevailing training method to encourage sparsity also introduces biases that harm the accuracy of SAE reconstructions, as the loss can be decreased by trading-off some reconstruction accuracy for lower L1. In this paper, we introduce a modification to the baseline SAE architecture – a *Gated SAE* – along with an accompanying loss function, which partially overcomes these limitations. Our key insight is to use separate affine transformations for (a) determining which dictionary elements to use in a reconstruction and (b) estimating the coefficients of active elements, and to apply the sparsity penalty only to the former task. We share a subset of weights between these transformations to avoid significantly

---

[*]: Joint contribution. [†]: Core infrastructure contributor. Correspondence: `srajamanoharan@google.com` and `neelnanda@google.com`.

38th Conference on Neural Information Processing Systems (NeurIPS 2024).

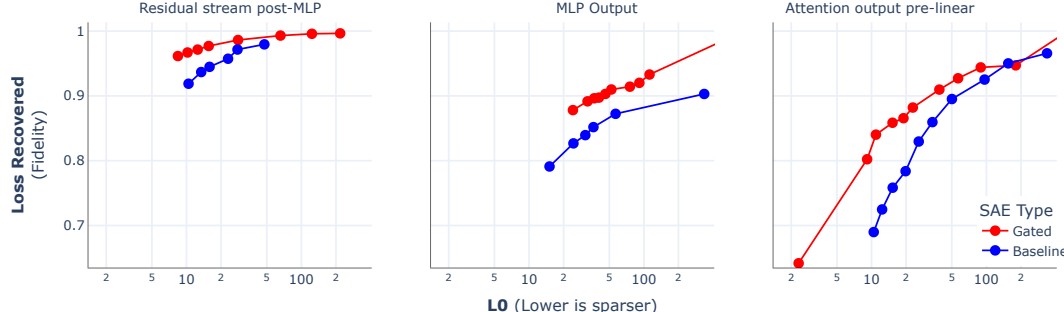

Figure 1: Gated SAEs consistently offer improved reconstruction fidelity for a given level of sparsity compared to prevailing (baseline) approaches. These plots compare Gated SAEs to baseline SAEs at Layer 20 in Gemma-7B. Gated SAEs' dictionaries are of size $2^{17} \approx 131k$ whereas baseline dictionaries are 50% larger, so that both types are trained with equal compute. This performance improvement holds in layers throughout GELU-1L, Pythia-2.8B and Gemma-7B (see Appendix E).

increasing the parameter count and inference-time compute requirements of a Gated SAE compared to a baseline SAE of equivalent width.[2]

We evaluate Gated SAEs on multiple models: a one layer GELU activation language model [28], Pythia-2.8B [3] and Gemma-7B [18], and on multiple sites within models: MLP layer outputs, attention layer outputs, and residual stream activations. Across these models and sites, we find Gated SAEs to be a Pareto improvement over baseline SAEs holding training compute fixed (Fig. 1): they yield sparser decompositions at any desired level of reconstruction fidelity. We also conduct further follow up ablations and investigations on a subset of these models and sites to better understand the differences between Gated SAEs and baseline SAEs.

Overall, the key contributions of this work are that we:

1. Introduce the Gated SAE, a modification to the standard SAE architecture that decouples detection of which features are present from estimating their magnitudes (Section 3.2);
2. Show that Gated SAEs Pareto improve the sparsity and reconstruction fidelity trade-off compared to baseline SAEs (Section 4.1);
3. Confirm that Gated SAEs overcome shrinkage while outperforming other methods that also address this problem (Section 5.2);
4. Provide evidence from a small double-blind study that Gated SAE features are comparably interpretable to baseline SAE features (Section 4.2).

## 2 Preliminaries

In this section we summarise the concepts and notation necessary to understand existing SAE architectures and training methods following Bricken et al. [8], which we call the *baseline SAE*. We define Gated SAEs in Section 3.2.

As motivated in Section 1, we wish to decompose a model's activation $\mathbf{x} \in \mathbb{R}^n$ into a sparse, linear combination of feature directions:

$$\mathbf{x} \approx \mathbf{x}_0 + \sum_{i=1}^{M} f_i(\mathbf{x})\mathbf{d}_i, \tag{1}$$

where $\mathbf{d}_i$ are dictionary of $M \gg n$ latent unit-norm *feature directions*, and the sparse coefficients $f_i(\mathbf{x}) \geq 0$ are the corresponding *feature activations* for $\mathbf{x}$.[3] The right-hand side of Eq. (1) naturally has the structure of an autoencoder: an input activation $\mathbf{x}$ is encoded into a (sparse) feature activations vector $\mathbf{f}(\mathbf{x}) \in \mathbb{R}^M$, which in turn is linearly decoded to reconstruct $\mathbf{x}$.

---

[2]Although due to an auxiliary loss term, computing the Gated SAE loss for training purposes does require 50% more compute than computing the loss for a matched-width baseline SAE.

[3]In this work, we use the term *feature* to refer only to the *learned features* of SAEs, i.e. the overcomplete basis directions that are linearly combined to produce reconstructions. In particular, *learned features* are always linear and not necessarily interpretable.

**Baseline architecture** Using this correspondence, Bricken et al. [8] and subsequent works attempt to learn a suitable sparse decomposition by parameterizing a single-layer autoencoder $(\mathbf{f}, \hat{\mathbf{x}})$ defined by:

$$\mathbf{f}(\mathbf{x}) := \mathrm{ReLU}\left(\mathbf{W}_{\mathrm{enc}}\left(\mathbf{x} - \mathbf{b}_{\mathrm{dec}}\right) + \mathbf{b}_{\mathrm{enc}}\right) \tag{2}$$

$$\hat{\mathbf{x}}(\mathbf{f}) := \mathbf{W}_{\mathrm{dec}}\mathbf{f} + \mathbf{b}_{\mathrm{dec}} \tag{3}$$

and training it using gradient descent to reconstruct samples $\mathbf{x} \sim \mathcal{D}$ from a large dataset $\mathcal{D}$ of activations collected from a single site and layer of a trained language model, constraining the hidden representation $\mathbf{f}$ to be sparse. Once the sparse autoencoder has been trained, we obtain a decomposition of the form of Eq. (1) by identifying the (suitably normalised) columns of the decoder weight matrix $\mathbf{W}_{\mathrm{dec}} \in \mathbb{R}^{M \times n}$ with the dictionary of feature directions $\mathbf{d_i}$, the decoder bias $\mathbf{b}_{\mathrm{dec}} \in \mathbb{R}^n$ with the centering term $\mathbf{x}_0$, and the (suitably normalised) entries of the latent representation $\mathbf{f}(\mathbf{x}) \in \mathbb{R}^M$ with the feature activations $f_i(\mathbf{x})$.

**Baseline training methodology** To train sparse autoencoders, Bricken et al. [8] use a loss function with two terms that respectively encourage faithful reconstruction and sparsity:[4]

$$\mathcal{L}(\mathbf{x}) := \|\mathbf{x} - \hat{\mathbf{x}}(\mathbf{f}(\mathbf{x}))\|_2^2 + \lambda \|\mathbf{f}(\mathbf{x})\|_1 . \tag{4}$$

Since it is possible to arbitrarily reduce the L1 sparsity loss term without affecting reconstructions or sparsity by simply scaling down encoder outputs and scaling up the norm of the decoder weights, it is important to constrain the norms of the columns of $\mathbf{W}_{\mathrm{dec}}$ during training. Following Bricken et al. [8], we constrain norms to one. See Appendix G for further details on SAE training.

**Evaluating SAEs** Two metrics are primarily used to get a sense of SAE quality [8]: *L0*, a measure of SAE sparsity, and *loss recovered*, a measure of SAE reconstruction fidelity. L0 measures the average number of features used by a SAE to reconstruct input activations. Loss recovered is a normalised measure of the increase induced in a LM's cross entropy loss when we replace its original activations with the corresponding SAE reconstructions during the model's forward pass. Both these metrics are formally defined in Appendix B, where we also discuss shortcomings of and alternatives to the loss recovered metric as it is defined in Bricken et al. [8]. Since it is possible for SAEs to score well on these metrics and still fail to be useful for interpretability-related tasks [47], we perform manual analysis of SAE interpretability in Section 4.2.

## 3 Gated SAEs

### 3.1 Motivation

The intention behind how SAEs are trained is to maximise reconstruction fidelity at a given level of sparsity, as measured by L0, although in practice we optimize a mixture of reconstruction fidelity and L1 regularization. This difference is a source of unwanted bias in the training of a sparse autoencoder: for any fixed level of sparsity, a trained SAE can achieve lower loss (as defined in Eq. (4)) by trading off a little reconstruction fidelity to perform better on the L1 sparsity penalty.

The clearest consequence of this bias is *shrinkage* [51]. Holding the decoder $\hat{\mathbf{x}}(\bullet)$ fixed, the L1 penalty pushes feature activations $\mathbf{f}(\mathbf{x})$ towards zero, while the reconstruction loss pushes $\mathbf{f}(\mathbf{x})$ high enough to produce an accurate reconstruction. Thus, the optimal value falls somewhere in between, and as a result the SAE systematically underestimates the magnitude of feature activations, without necessarily providing any compensatory benefit for sparsity.[5]

How can we reduce the bias introduced by the L1 penalty? The output of the encoder $\mathbf{f}(\mathbf{x})$ of a baseline SAE (Section 2) has two roles:

1. It *detects* which features are active (according to whether the outputs are zero or strictly positive). For this role, the L1 penalty is necessary to ensure the decomposition is sparse.

---

[4]Note that we cannot directly optimize the L0 norm (i.e. the number of active features) since this is not a differentiable function. We do however use the L0 norm to evaluate SAE sparsity.

[5]Conversely, rescaling the shrunk feature activations [51] is not necessarily enough to overcome the bias induced by by L1 penalty: a SAE trained with the L1 penalty could have learnt sub-optimal encoder and decoder directions that are not improved by such a fix. In Section 5.2 and Fig. 7 we provide empirical evidence that this is true in practice.

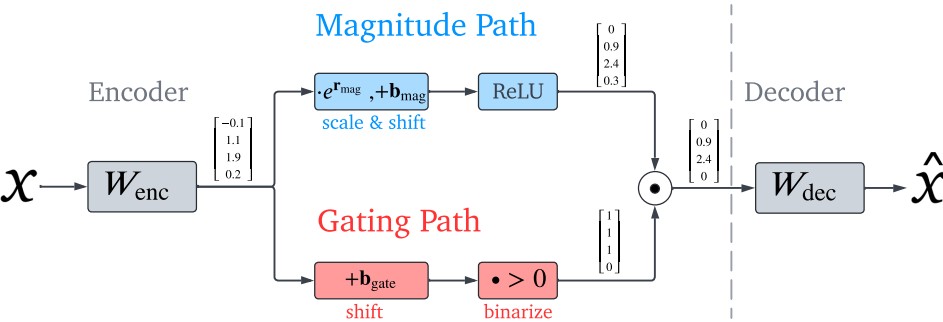

Figure 2: The Gated SAE architecture with weight sharing between the gating and magnitude paths, shown with an example input. See Appendix J for a pseudo-code implementation.

2. It *estimates* the magnitudes of active features. For this role, the L1 penalty is a source of unwanted bias.

If we could separate out these two functions of the SAE encoder, we could design a training loss that narrows down the scope of SAE parameters that are affected (and therefore to some extent biased) by the L1 sparsity penalty to precisely those parameters that are involved in feature detection, minimising its impact on parameters used in feature magnitude estimation.

### 3.2 Gated SAEs

**Architecture**   How should we modify the baseline SAE encoder to achieve this separation of concerns? Our solution is to replace the single-layer ReLU encoder of a baseline SAE with a *gated* ReLU encoder. Taking inspiration from Gated Linear Units [43, 12], we define the gated encoder as

$$\tilde{\mathbf{f}}(\mathbf{x}) := \underbrace{\mathbb{1}[\overbrace{(\mathbf{W}_{\text{gate}}(\mathbf{x} - \mathbf{b}_{\text{dec}}) + \mathbf{b}_{\text{gate}})}^{\boldsymbol{\pi}_{\text{gate}}(\mathbf{x})} > \mathbf{0}]}_{\mathbf{f}_{\text{gate}}(\mathbf{x})} \odot \underbrace{\text{ReLU}(\mathbf{W}_{\text{mag}}(\mathbf{x} - \mathbf{b}_{\text{dec}}) + \mathbf{b}_{\text{mag}})}_{\mathbf{f}_{\text{mag}}(\mathbf{x})}, \qquad (5)$$

where $\mathbb{1}[\bullet > \mathbf{0}]$ is the (pointwise) Heaviside step function and $\odot$ denotes elementwise multiplication. Here, $\mathbf{f}_{\text{gate}}$ determines which features are deemed to be active, while $\mathbf{f}_{\text{mag}}$ estimates feature activation magnitudes (which only matter for features that have been deemed to be active); $\boldsymbol{\pi}_{\text{gate}}(\mathbf{x})$ are the $\mathbf{f}_{\text{gate}}$ sub-layer's pre-activations, which are used in the gated SAE loss, defined below.

**Training**   A naive guess at a loss function for training Gated SAEs would be to replace the sparsity penalty in Eq. (4) with the L1 norm of $\mathbf{f}_{\text{gate}}(\mathbf{x})$. Unfortunately, due to the Heaviside step activation function in $\mathbf{f}_{\text{gate}}$, no gradients would propagate to $\mathbf{W}_{\text{gate}}$ and $\mathbf{b}_{\text{gate}}$. To mitigate this, we instead apply the L1 norm to the positive parts of the preactivation, $\text{ReLU}(\boldsymbol{\pi}_{\text{gate}}(\mathbf{x}))$. To ensure $\mathbf{f}_{\text{gate}}$ aids reconstruction by detecting active features, we add an auxiliary task requiring that these same rectified preactivations can be used by the decoder to produce a good reconstruction:

$$\mathcal{L}_{\text{gated}}(\mathbf{x}) := \underbrace{\left\| \mathbf{x} - \hat{\mathbf{x}}\left(\tilde{\mathbf{f}}(\mathbf{x})\right) \right\|_2^2}_{\mathcal{L}_{\text{reconstruct}}} + \underbrace{\lambda \left\| \text{ReLU}(\boldsymbol{\pi}_{\text{gate}}(\mathbf{x})) \right\|_1}_{\mathcal{L}_{\text{sparsity}}} + \underbrace{\left\| \mathbf{x} - \hat{\mathbf{x}}_{\text{frozen}}\left(\text{ReLU}\left(\boldsymbol{\pi}_{\text{gate}}(\mathbf{x})\right)\right) \right\|_2^2}_{\mathcal{L}_{\text{aux}}} \quad (6)$$

where $\hat{\mathbf{x}}_{\text{frozen}}$ is a frozen copy of the decoder, $\hat{\mathbf{x}}_{\text{frozen}}(\mathbf{f}) := \mathbf{W}_{\text{dec}}^{\text{copy}}\mathbf{f} + \mathbf{b}_{\text{dec}}^{\text{copy}}$, to ensure that gradients from $\mathcal{L}_{\text{aux}}$ do not propagate back to $\mathbf{W}_{\text{dec}}$ or $\mathbf{b}_{\text{dec}}$ . This can be implemented by stop gradient operations rather than creating copies. See Appendix J for pseudo-code for the forward pass and loss function.

To calculate this loss (or its gradient), we have to run the decoder twice: once to perform the main reconstruction for $\mathcal{L}_{\text{reconstruct}}$ and once to perform the auxiliary reconstruction for $\mathcal{L}_{\text{aux}}$. This leads to a 50% increase in the compute required to perform a training update step. However, the increase in overall training time is typically much less, as in our experience much of the training wall clock time goes to generating language model activations (if these are being generated on the fly) or disk I/O (if training on saved activations).

**Parameter reduction through weight-tying**    Naively, we appear to have doubled the number of parameters in the encoder, increasing the total number of parameters by 50% with respect to baseline SAEs. We mitigate this through weight sharing: we parameterize these layers so that the two layers share the same projection directions, but allow the norms of these directions as well as the layer biases to differ. Concretely, we define $\mathbf{W}_{\text{mag}}$ in terms of $\mathbf{W}_{\text{gate}}$ and an additional vector-valued rescaling parameter $\mathbf{r}_{\text{mag}} \in \mathbb{R}^M$ as follows:

$$(\mathbf{W}_{\text{mag}})_{ij} := (\exp(\mathbf{r}_{\text{mag}}))_i \cdot (\mathbf{W}_{\text{gate}})_{ij}. \tag{7}$$

See Fig. 2 for an illustration of the tied-weight Gated SAEs architecture. With this weight tying scheme, the Gated SAE has only $2 \times M$ more parameters than a baseline SAE. In Section 5.1, we show that this weight tying scheme does not harm performance.

With tied weights, the gated encoder can be reinterpreted as a single-layer linear encoder with a non-standard and discontinuous "JumpReLU" activation function [17], $\sigma_\theta(z)$, illustrated in Fig. 12. To be precise, using the weight tying scheme of Eq. (7), $\tilde{\mathbf{f}}(\mathbf{x})$ can be re-expressed as $\tilde{\mathbf{f}}(\mathbf{x}) = \sigma_{\boldsymbol{\theta}}(\mathbf{W}_{\text{mag}} \cdot \mathbf{x} + \mathbf{b}_{\text{mag}})$, with the JumpReLU gap given by $\boldsymbol{\theta} = \mathbf{b}_{\text{mag}} - e^{\mathbf{r}_{\text{mag}}} \odot \mathbf{b}_{\text{gate}}$; see Appendix H for an explanation. We think this is a useful intuition for reasoning about how Gated SAEs reconstruct activations in practice.

## 4    Evaluating Gated SAEs

In this section we benchmark Gated SAEs against baseline SAEs across a large variety of models and at different sites. We show that they produce more faithful reconstructions at equal sparsity and that they resolve shrinkage. Through a double-blind manual interpretability study, we find that Gated SAEs produce features that are similarly interpretable to baseline SAE features.

### 4.1    Benchmarking Gated SAEs

**Methodology**    We trained a suite of Gated and baseline SAEs, a family of each type to reconstruct each of the following activations:

1. The MLP neuron activations in GELU-1L, which is the closest direct comparison to Bricken et al. [8];
2. The MLP outputs, attention layer outputs (taken pre-$W_O$ [21]) and residual stream activations in 5 different layers throughout Pythia-2.8B and four different layers in the Gemma-7B base model.

For each model and reconstruction site, we trained multiple SAEs using different values of $\lambda$ (and therefore L0), allowing us to compare the Pareto frontiers of L0 and loss recovered between Gated and baseline SAEs. We also use the *relative reconstruction bias* metric, $\gamma$, defined in Appendix C to measure shrinkage in our trained SAEs. This metric measures the relative bias in the norm of an SAE's reconstructions; unbiased SAEs obtain $\gamma = 1$, whereas SAEs affected by shrinkage (which causes reconstruction norms to be systematically too small) have $\gamma < 1$.

Since Gated SAEs require at most $1.5\times$ more compute to train than regular SAEs (Section 3.2) of the same width, we compare Gated SAEs to baseline SAEs that have a 50% larger dictionary (hidden dimension $M$) to ensure fair comparison in our evaluations.[6]

**Results**    We plot sparsity against reconstruction fidelity for SAEs with different values of $\lambda$. Higher $\lambda$ corresponds to increased sparsity and worse reconstruction, so as in Bricken et al. [8] we observe a Pareto frontier of possible trade-offs. We plot Pareto curves for GELU-1L in Fig. 3a and Pythia-2.8B and Gemma-7B in Appendix E. At all sites tested, Gated SAEs are a Pareto improvement over regular SAEs: they provide better reconstruction fidelity at any fixed level of sparsity.[7] For some sites in Pythia-2.8B and Gemma-7B, loss recovered does not monotonically increase with L0; we attribute this to difficulties training SAEs (Appendix G.1.3). Finally, full tables of results for Pythia and Gemma can be found in Appendix K.

---

[6]Since wider SAEs provide better reconstructions (all else being equal), the gap between Gated SAEs' and baseline SAEs' performance is even wider when we use baseline SAEs with equal width in the comparison. This can be seen in the difference between the "$1.5\times$ width" and "equal width" baseline curves in Fig. 5.

[7]Although both Gated and baseline SAEs have loss recovered tending to one for high enough L0.

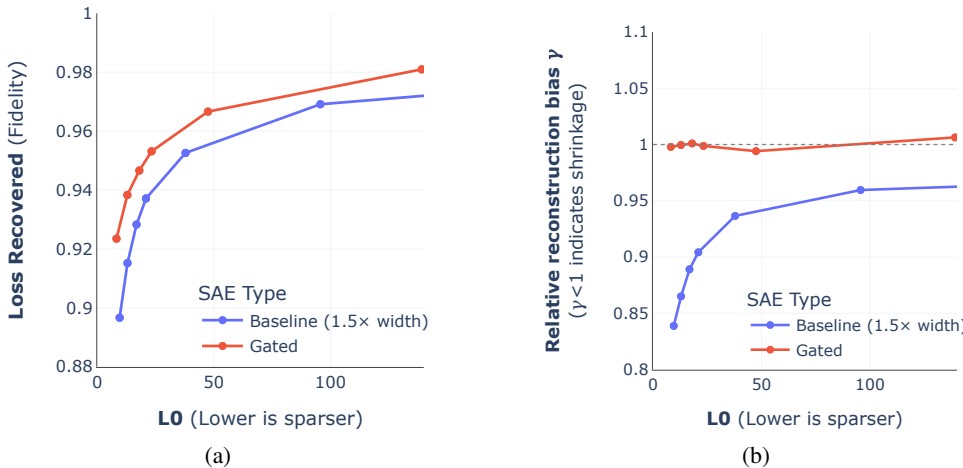

(a)            (b)

Figure 3: (a) Gated SAEs offer better reconstruction fidelity (as measured by loss recovered) at any given level of feature sparsity (as measured by L0); (b) Gated SAEs address shrinkage. These plots compare Gated and baseline SAEs trained on GELU-1L neuron activations; see Appendix E for comparisons on Pythia-2.8B and Gemma-7B.

As shown in Fig. 3b, Gated SAEs' reconstructions are unbiased, with $\gamma \approx 1$, whereas baseline SAEs exhibit shrinkage ($\gamma < 1$), with the impact of shrinkage getting worse as the L1 coefficient $\lambda$ increases (and L0 consequently decreases). Fig. 10 shows that this result generalizes to Pythia-2.8B.

## 4.2 Interpretability

Although Gated SAEs provide more faithful reconstructions than baselines at equal sparsity, it does not necessarily follow that these reconstructions are better suited to downstream interpretability-related tasks. Currently, there is no consensus on how to systematically assess the degree to which a SAE's features are useful for downstream tasks, but a plausible proxy is to assess the extent to which these features are human interpretable [8]. Therefore, to gain a more qualitative understanding of the differences between their learned features, we conduct a blinded human study in which we rate and compare the interpretability of randomly sampled Gated and baseline SAE features.

**Methodology**  We study a variety of SAEs from different layers and sites. For Pythia-2.8B we had 5 raters, who each rated one feature from baseline and Gated SAEs trained on each (site, layer) pair from Fig. 8, for a total of 150 features. For Gemma-7B we had 7 raters; one rated 2 features each, and the rest 1 feature each, from baseline or Gated SAEs trained on each (site, layer) pair from Fig. 9, for a total of 192 features.

For each model, raters are shown the features in random order, without revealing which SAE, site, or layer they came from.[8] To assess a feature, the rater decides whether there is an explanation of the feature's behavior, in particular for its highest activating examples. The rater then enters that explanation (if applicable) and selects whether the feature is interpretable ('Yes'), uninterpretable ('No') or maybe interpretable ('Maybe'). All raters are either authors of this paper or colleagues, who have prior experience interpreting SAE features. As an interface we use an open source SAE visualizer library [27]; representative screenshots of the dashboards produced by this library are shown in Fig. 14.

**Results & analysis**  Fig. 4 shows interpretability rating distributions by SAE type and LM, marginalising over layers, sites and raters.[9] To compare the interpretability of baseline and Gated SAEs, we first pair our datapoints according to all covariates (model, layer, site, rater); this lets us

---

[8]Although due to a debugging issue, Gemma-7B attention SAEs were rated separately, so raters were not blind to that.

[9]95% error bars were obtained by modelling each frequency shown as binomial, with $p$ set to the sample frequency, and calculating the 2.5% and 97.5% quantiles.

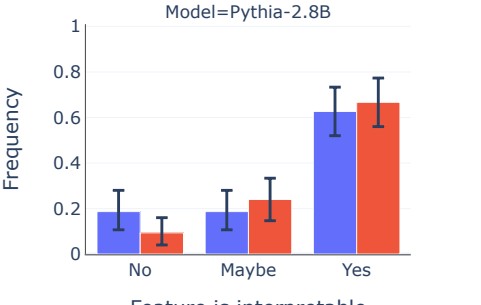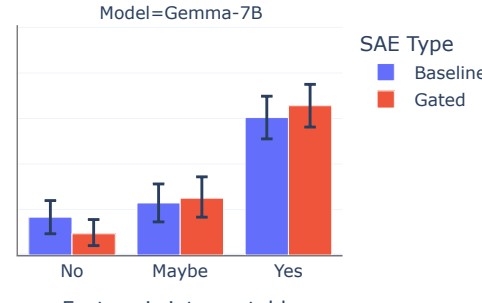

Figure 4: Proportions of SAE features rated as interpretable / uninterpretable / maybe interpretable by SAE type (Gated or baseline) and language model. Gated and baseline SAEs are similarly interpretable, with a mean difference (in favor of Gated SAEs) of 0.13 (95% CI $[0, 0.26]$) after aggregating ratings for both models.

control for all of them without making any parametric assumptions, and thus reduces variance in the comparison. We then measure the mean difference between baseline and Gated labels, where we count 'No' as 0, 'Maybe' as 1, and 'Yes' as 2, and compute a 90% BCa bootstrap confidence interval. Thus we find that the mean difference in label scores is 0.13 (90% CI $[0, 0.26]$) in favour of Gated SAEs, breaking down to mean difference CIs of $[-0.07, 0.33]$ and $[-0.04, 0.29]$ on just the Pythia-2.8B data and Gemma-7B data respectively. Since our central estimate for the mean difference in scores is positive, we also test the hypothesis that Gated SAEs may be *more* interpretable than baseline SAEs. However, a one-sided Wilcoxon-Pratt signed-rank test on the paired scores does not reject the null hypothesis that they are equally interpretable ($p = 0.06$). The contingency tables used for these results are shown in Fig. 13. The overall conclusion is that Gated SAE features are similarly interpretable to baseline SAE features, while also providing better reconstruction fidelity (at fixed sparsity), as shown in the previous section. We provide more analysis of how these break down by site and layer in Appendix I.

## 5 Why do Gated SAEs improve SAE training?

### 5.1 Ablation study

In this section, we vary several parts of the Gated SAE training methodology to gain insight into which aspects of the training drive the observed improvement in performance. Gated SAEs differ from baseline SAEs in many respects, making it easy to incorrectly attribute the performance gains to spurious details without a careful ablation study. Fig. 5a shows Pareto frontiers for these variations; below we describe each variation in turn and discuss our interpretation of the results.

**Unfreeze decoder**: Here we unfreeze the decoder weights in $\mathcal{L}_{\text{aux}}$ – i.e. allow this auxiliary task to update the decoder weights in addition to training $\mathbf{f}_{\text{gate}}$'s parameters. Although this (slightly) simplifies the loss, there is a reduction in performance, suggesting that it is beneficial to limit the impact of the L1 sparsity penalty to just those parameters in the SAE that need it – i.e. those used to detect which features are active.

**No $\mathbf{r}_{\text{mag}}$**: Here we remove the $\mathbf{r}_{\text{mag}}$ scaling parameter in Eq. (7), effectively setting it to zero, further tying $\mathbf{f}_{\text{gate}}$'s and $\mathbf{f}_{\text{mag}}$'s parameters together. With this change, the two encoder sublayers' preactivations can at most differ by an elementwise shift.[10] There is a slight drop in performance, suggesting $\mathbf{r}_{\text{mag}}$ contributes somewhat to the improved performance of the Gated SAE.

**Untied encoders**: Here we check whether our choice to share the majority of parameters between the two encoders has meaningfully hurt performance, by training Gated SAEs with gating and ReLU encoder parameters completely untied. Despite the greater expressive power of an untied encoder, we see no improvement in performance – in fact a slight deterioration. This suggests our tying scheme (Eq. (7)) – where encoder directions are shared, but magnitudes and biases aren't – is effec-

---

[10]Because the two biases $\mathbf{b}_{\text{gate}}$ and $\mathbf{b}_{\text{mag}}$ can still differ.

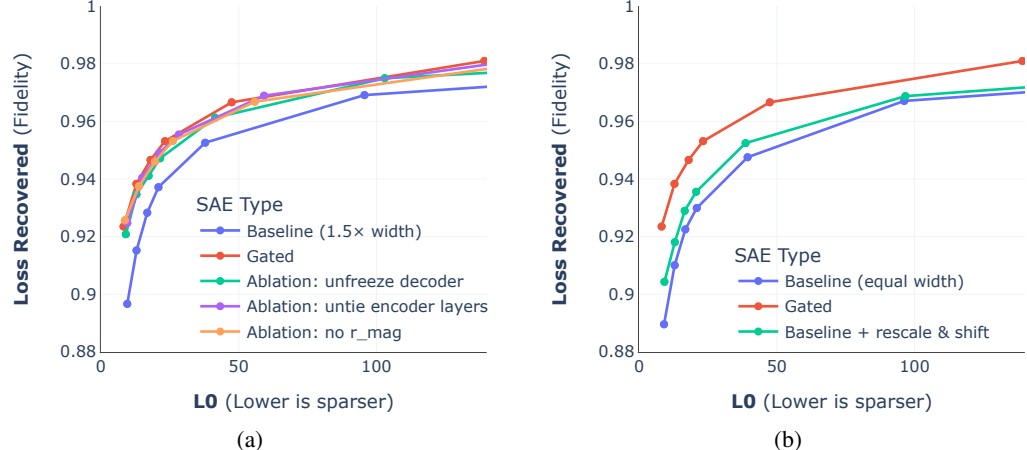

Figure 5: (a) Our ablation study on GELU-1L MLP neuron activations indicates: (i) the importance of freezing the decoder in the auxiliary task $\mathcal{L}_{aux}$ used to train $\mathbf{f}_{gate}$'s parameters; (ii) tying encoder weights according to Eq. (7) is slightly beneficial for performance (in addition to yielding a significant reduction in parameter count and inference compute); (iii) further simplifying the encoder weight tying scheme in Eq. (7) by removing $\mathbf{r}_{mag}$ is mildly harmful to performance. (b) Evidence from GELU-1L that the performance improvement of gated SAEs does not solely arise from addressing shrinkage (systematic underestimation of latent feature activations): taking a frozen baseline SAE's parameters and learning $\mathbf{r}_{mag}$ and $\mathbf{b}_{mag}$ parameters on top of them (green line) does successfully resolve shrinkage, by decoupling feature magnitude estimation from active feature detection; however, it explains only a small part of the performance increase of gated SAEs (red line) over baseline SAEs (blue line).

tive at capturing the advantages of using a gated SAE while avoiding the 50% increase in parameter count and inference-time compute of using an untied SAE.

## 5.2   Is it sufficient to just address shrinkage?

As explained in Section 3.1, SAEs trained with the baseline architecture and L1 loss systematically underestimate the magnitudes of latent features' activations (i.e. shrinkage). Gated SAEs, through modifications to their architecture and loss function, overcome these limitations.

It is natural to ask to what extent the performance improvement of Gated SAEs is solely attributable to addressing shrinkage. Although addressing shrinkage would – all else staying equal – improve reconstruction fidelity, it is not the only way to improve SAEs' performance: for example, Gated SAEs could also improve upon baseline SAEs by learning better encoder directions (for estimating when features are active and their magnitudes) or by learning better decoder directions (i.e. better dictionaries for reconstructing activations).

Here we try to answer this question by comparing Gated SAEs trained as described in Section 3.2 with an alternative (architecturally equivalent) approach that also addresses shrinkage, but in a way that uses frozen encoder and decoder directions from a baseline SAE of equal dictionary size.[11] Any performance improvement over baseline SAEs obtained by this alternative approach (which we dub "baseline + rescale & shift") can only be due to better estimations of active feature magnitudes, since by construction an SAE parameterized by "baseline + rescale & shift" shares the same encoder and decoder directions as a baseline SAE.

As shown in Fig. 5b, although resolving shrinkage only ("baseline + rescale & shift") does improvement baseline SAEs' performance a little, a significant gap remains with respect to the performance of Gated SAEs. This suggests that the benefit of the gated architecture and loss comes from learning better encoder and decoder directions, not just from overcoming shrinkage. In Appendix D we ex-

---

[11]Concretely, we do this by training baseline SAEs, freezing their weights, and then learning additional rescale and shift parameters (similar to Wright and Sharkey [51]) to be applied to the (frozen) encoder pre-activations before estimating feature magnitudes.

plore further how Gated and baseline SAEs' decoders differ by replacing their respective encoders with an optimization algorithm at inference time.

## 6 Related work

**Mechanistic interpretability**   Recent work in mechanistic interpretability has found recurring components in small and large LMs [38], identified computational subgraphs that carry out specific tasks in small LMs (circuits; [50]) and reverse-engineered how toy tasks are carried out in small transformers [30]. A central difficulty in this kind of work is choosing the right units of analysis. Sparse linear features have been identified as a promising candidate in prior work [52, 46]. The superposition hypothesis outlined by Elhage et al. [16] also provided a theoretical basis for this theory, sparking a new interest in using SAEs specifically to learn a feature basis [42, 8, 11, 21, 22, 4], as well as using SAEs directly for circuit analysis [26]. Other work has drawn awareness to issues or drawbacks with SAE training for this purpose, some of which our paper mitigates. Wright and Sharkey [51] raised awareness of shrinkage and proposed addressing this via fine-tuning. Gated SAEs, as discussed, resolve shrinkage during training. [35, 47, 2, 36] have also proposed general SAE training methodology improvements, which are mostly orthogonal to the architectural changes discussed in this work. In parallel work, Taggart [45] finds early improvements using a Jump ReLU [17], but with a different loss function, and without addressing the problems of the L1 penalty.

**Classical dictionary learning**   Research into the general problem of sparse dictionary learning precedes transformers, and even deep learning. For example, sparse coding [13] studies how discrete and continuous representations can involve more representations than basis vectors, and sparse representations are also studied in neuroscience [48, 37]. One dictionary learning algorithm, k-SVD [1] also uses two stages to learn a dictionary like Gated SAEs. Although classical dictionary learning algorithms can be more powerful than SAEs (Appendix D), they are less suited for downstream uses like weights-based circuit analysis or attribution patching [44, 24], because they typically use an iterative algorithm to decompose activations, whereas SAEs make feature extraction explicit via the encoder. Bricken et al. [8] have also argued that classical algorithms may be 'too strong', in the sense they may learn features the LM itself could not access, whereas SAEs uses components similar to a LM's MLP layer to decompose activations.

## 7 Conclusion

In this work we introduced Gated SAEs which are a Pareto improvement in terms of reconstruction quality and sparsity compared to baseline SAEs (Section 4.1), and are comparably interpretable (Section 4.2). We showed via an ablation study that every key part of the Gated SAE methodology was necessary for strong performance (Section 5.1). This represents significant progress on improving Dictionary Learning on LMs – at many sites, Gated SAEs require half the L0 to achieve the same loss recovered (Fig. 8). This is likely to improve work that uses SAEs to steer language models [31], interpret circuits [26], or understand LM components across the full distribution [8].

**Limitations & future work**. Our benchmarking study focused on GELU-1L and models in the Pythia and Gemma families. It is therefore not certain that these results will generalise to other model families. On the other hand, the theoretical underpinnings of the Gated SAE architecture (Section 3) make no assumptions about LM architecture, suggesting Gated SAEs should be a Pareto improvement more generally. While we have confirmed that Gated SAE features are comparably interpretable to baseline SAE features, it does not necessarily follow that Gated SAE decompositions are equally useful for mechanistic interpretability. It is certainly possible that human interpretability of SAE features is only weakly correlated with either: (i) identification of the causally meaningful directions in a LM's activations; or (ii) usefulness on downstream tasks like circuit analysis or steering. A framework for scalably and objectively evaluating the usefulness of SAE decompositions (gated or otherwise) is still in its early stages [25] and further progress in this area would be highly valuable. It is plausible that some of the performance gap between Gated and baseline SAEs could be closed by inexpensive inference-time interventions that prune the many low activating features that tend to appear in baseline SAEs, mimicking Gated SAEs' thresholding mechanism. Finally, we would be most excited to see progress on using dictionary learning techniques to further inter-

pretability in general, such as to improve circuit finding [10, 26] or steering [49] in language models, and hope that Gated SAEs can serve to accelerate such work.

## Acknowledgements

We would like to thank Romeo Valentin for conversations that got us thinking about k-SVD in the context of SAEs, which inspired part of our work. Additionally, we are grateful for Vladimir Mikulik's detailed feedback on a draft of this work which greatly improved our presentation, and Nicholas Sonnerat's work on our codebase and help with feature labelling. We would also like to thank Glen Taggart who found in parallel work [45] that a similar method gave improvements to SAE training, helping give us more confidence in our results. We are grateful to Sam Marks for pointing out an error in the derivation of relative reconstruction bias in an earlier version of this paper and Leo Gao and Gonçalo Paulo for discussions. Finally, we thank our anonymous reviewers for their valuable feedback, which helped improve the quality of this paper.

## Author contributions

Senthooran Rajamanoharan developed the Gated SAE architecture and training methodology, inspired by discussions with Lewis Smith on the topic of shrinkage. Arthur Conmy and Senthooran Rajamanoharan performed the mainline experiments in Section 4 and Section 5 and led the writing of all sections of the paper. Tom Lieberum implemented the manual interpretability study of Section 4.2, which was designed and analysed by János Kramár. Tom Lieberum also created Fig. 2 and Lewis Smith contributed Appendix D. Our SAE codebase was designed by Vikrant Varma who implemented it with Tom Lieberum, and was scaled to Gemma by Arthur Conmy, with contributions from Senthooran Rajamanoharan and Lewis Smith. János Kramár built most of our underlying interpretability infrastructure. Rohin Shah and Neel Nanda edited the manuscript and provided leadership and advice throughout the project.

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

# Appendix

## A   Impact statement

This work introduces a method to obtain higher fidelity sparse decompositions of LM activations, under the hypothesis that progress in this area will ultimately help us understand the representations used by LMs. If successful, this could lead to greater understanding of how LMs complete tasks and novel mechanisms for controlling their behavior. Greater understanding and control could be put to beneficial uses such as mitigating the harms caused by current and future models, although bad actors could also misuse these tools, for example to circumvent safety training and steer models towards harmful behaviors. Currently, the SAE research program is in its early stages. For any potential misuse of SAEs, there is typically a more practical and effective way to achieve the same end using existing tooling, e.g. fine tuning or activation editing. Therefore, we see negligible negative societal impact in the short term. Longer term, advances in LM interpretability and control pose similar benefits and risks to advances in AI capabilities in general.

## B   Metrics for evaluating SAEs

SAEs are expected to decompose input activations sparsely, and yet in a manner that allows for faithful reconstruction. L0 and loss recovered are two metrics typically used [8] to measure sparsity and reconstruction fidelity respectively. These are defined as follows:

- The **L0** of a SAE is defined by the average number of active features on a given input, i.e $\mathbb{E}_{\mathbf{x} \sim \mathcal{D}} \|\mathbf{f}(\mathbf{x})\|_0$.
- The **loss recovered** of a SAE is calculated from the average cross-entropy loss of the language model on an evaluation dataset, when the SAE's reconstructions are spliced into it. If we denote by $\mathrm{CE}(\phi)$ the average loss of the language model when we splice in a function $\phi : \mathbb{R}^n \to \mathbb{R}^n$ at the SAE's site during the model's forward pass, then loss recovered is

$$1 - \frac{\mathrm{CE}(\hat{\mathbf{x}} \circ \mathbf{f}) - \mathrm{CE}(\mathrm{Id})}{\mathrm{CE}(\zeta) - \mathrm{CE}(\mathrm{Id})}, \tag{8}$$

where $\hat{\mathbf{x}} \circ \mathbf{f}$ is the autoencoder function, $\zeta : \mathbf{x} \mapsto \mathbf{0}$ the zero-ablation function and Id : $\mathbf{x} \mapsto \mathbf{x}$ the identity function. According to this definition, a SAE that always outputs the zero vector as its reconstruction would get a loss recovered of 0%, whereas a SAE that reconstructs its inputs perfectly would get a loss recovered of 100%.

### B.1   Issues with the loss recovered metric

In this paper, we have used loss recovered as defined in Bricken et al. [8] to measure reconstruction fidelity. However, there are deficiencies with this metric:

- Firstly, zero-ablation is arguably too poor a baseline for defining the zero-point of this metric and mean-ablation is better justified. Using the mean-ablation function $\mu : \mathbf{x} \mapsto \mathbb{E}_{\mathbf{x}' \sim \mathcal{D}} \mathbf{x}'$, instead of $\zeta$ in the definition of loss recovered above would also have the benefit that SAEs' loss recovered would tend towards zero in the limit $L0 \to 0$, instead of tending to a positive value as it does when computing loss recovered using zero-ablation.
- Furthermore, the very fact we normalise the increase in the spliced LM's loss when computing loss recovered makes it difficult to compare the impact of splicing SAEs at different sub-layers of the model. For example, mean or zero-ablating the output of a MLP layer typically has a much milder impact on LM loss than mean or zero-ablating the residual stream, making the denominator in Eq. (8) smaller for MLP SAEs than for residual stream SAEs. So we unsurprisingly find that residual stream SAEs' loss recovered tend to be much higher than MLP or attention SAEs. This suggests that it may be more informative to report raw changes in cross-entropy loss ("delta LM loss") instead of using a normalised metric like loss recovered, since these are directly comparable across SAEs trained on different sub-layers of the same LM.

In practice however, both mean-ablated loss recovered and delta LM loss are related to zero-ablated loss recovered (the metric used in this paper) by an affine transformation. In other words, all the

loss recovered versus L0 figures in this paper would look identical if we had used one of these other metrics instead, with the only difference being the tick labels on the y-axis. Consequently, none of the conclusions we draw in this paper would be affected by using one of these other reconstruction fidelity metrics instead. Nevertheless, we draw the reader's attention to our subsequent work [41], which compares Gated SAEs to other SAE varieties adopting the delta LM loss metric, instead of loss recovered, for measuring reconstruction fidelity.[12]

## C   Measuring shrinkage

As described in Section 3.1, the L1 sparsity penalty used to train baseline SAEs causes feature activations to be systematically underestimated, a phenomenon called *shrinkage*. Since this in turn shrinks the reconstructions produced by the SAE decoder, we can observe the extent to which a trained SAE is affected by shrinkage by measuring the average norm of its reconstructions.

Concretely, the metric we use is the *relative reconstruction bias*,

$$\gamma := \arg\min_{\gamma'} \mathbb{E}_{\mathbf{x}\sim\mathcal{D}}\left[\|\hat{\mathbf{x}}_{\text{SAE}}(\mathbf{x})/\gamma' - \mathbf{x}\|_2^2\right], \tag{9}$$

i.e. $\gamma^{-1}$ is the optimum multiplicative factor by which an SAE's reconstructions should be rescaled in order to minimise the L2 reconstruction loss; $\gamma = 1$ for an unbiased SAE and $\gamma < 1$ when there's shrinkage.[13] Explicitly solving the optimization problem in Eq. (9), the relative reconstruction bias can be expressed analytically in terms of the mean SAE reconstruction loss, the mean squared norm of input activations and the mean squared norm of SAE reconstructions, making $\gamma$ easy to compute and track during training:

$$\gamma = \frac{\mathbb{E}_{\mathbf{x}\sim\mathcal{D}}\left[\|\hat{\mathbf{x}}_{\text{SAE}}(\mathbf{x})\|_2^2\right]}{\mathbb{E}_{\mathbf{x}\sim\mathcal{D}}\left[\hat{\mathbf{x}}_{\text{SAE}}(\mathbf{x})\cdot\mathbf{x}\right]} = \frac{2\,\mathbb{E}_{\mathbf{x}\sim\mathcal{D}}\left[\|\hat{\mathbf{x}}_{\text{SAE}}(\mathbf{x})\|_2^2\right]}{\mathbb{E}_{\mathbf{x}\sim\mathcal{D}}\left[\|\hat{\mathbf{x}}_{\text{SAE}}(\mathbf{x})\|_2^2\right] + \mathbb{E}_{\mathbf{x}\sim\mathcal{D}}\left[\|\mathbf{x}\|_2^2\right] - \mathbb{E}_{\mathbf{x}\sim\mathcal{D}}\left[\|\hat{\mathbf{x}}_{\text{SAE}}(\mathbf{x})-\mathbf{x}\|_2^2\right]}, \tag{10}$$

where the second equality makes use of the identity $2\mathbf{a}\cdot\mathbf{b} \equiv \|\mathbf{a}\|_2^2 + \|\mathbf{b}\|_2^2 - \|\mathbf{a}-\mathbf{b}\|_2^2$. Notice from the second expression for $\gamma$ that an unbiased reconstruction ($\gamma = 1$) therefore satisfies

$$\mathbb{E}_{\mathbf{x}\sim\mathcal{D}}\left[\|\hat{\mathbf{x}}_{\text{SAE}}(\mathbf{x})\|_2^2\right] = \mathbb{E}_{\mathbf{x}\sim\mathcal{D}}\left[\|\mathbf{x}\|_2^2\right] - \mathbb{E}_{\mathbf{x}\sim\mathcal{D}}\left[\|\hat{\mathbf{x}}_{\text{SAE}}(\mathbf{x})-\mathbf{x}\|_2^2\right].$$

In other words, an unbiased but imperfect SAE (i.e. one that has non-zero reconstruction loss) must have mean squared reconstruction norm that is strictly *less than* the mean squared norm of its inputs *even without shrinkage*. Shrinkage makes the mean squared reconstruction norm even smaller.

## D   Inference-time optimization

The task SAEs perform can be split into two sub-tasks: sparse coding, or learning a set of features from a dataset, and sparse approximation, where a given datapoint is approximated as a sparse linear combination of these features. The decoder weights are the set of learned features, and the mapping represented by the encoder is a sparse approximation algorithm. Formally, sparse approximation is the problem of finding a vector $\boldsymbol{\alpha}$ that minimises;

$$\boldsymbol{\alpha} = \arg\min \|\mathbf{x} - \mathbf{D}\boldsymbol{\alpha}\|_2^2 \quad s.t. \quad \|\boldsymbol{\alpha}\|_0 < \gamma \tag{11}$$

i.e. that best reconstructs the signal $\mathbf{x}$ as a linear combination of vectors in a dictionary $\mathbf{D}$, subject to a constraint on the L0 pseudo-norm on $\boldsymbol{\alpha}$. Sparse approximation is a well studied problem, and SAEs are a *weak* sparse approximation algorithm. SAEs, at least in the formulation conventional in dictionary learning for language models, in fact solve a slightly more restricted version of this problem where the weights $\boldsymbol{\alpha}$ on each feature are constrained to be non-negative, leading to the related problem

$$\boldsymbol{\alpha} = \arg\min \|\mathbf{x} - \mathbf{D}\boldsymbol{\alpha}\|_2^2 \quad s.t. \quad \|\boldsymbol{\alpha}\|_0 < \gamma, \boldsymbol{\alpha} > 0 \tag{12}$$

---

[12]We also provide in Table 1 cross entropy losses for the LMs used in our experiments, both with and without zero-ablation, which could in principle be used to translate the loss recovered results in this paper to delta LM loss.

[13]We have defined $\gamma$ this way round so that $\gamma < 1$ intuitively corresponds to shrinkage.

In this paper, we do not explore using more powerful algorithms for sparse coding. This is partly because we are using SAEs not just to recover *a* sparse reconstruction of activations of a LM; ideally we hope that the learned features will coincide with the linear representations actually used by the LM, under the superposition hypothesis. Prior work [8] has argued that SAEs are more likely to recover these due to the correspondence between the SAE encoder and the structure of the network itself; the argument is that it is implausible that the network can make use of features which can only be recovered from the vector via an iterative optimisation algorithm, whereas the structure of the SAE means that it can only find features whose presence can be predicted well by a simple linear mapping. Whether this is true remains, in our view, an important question for future work, but we do not address it in this paper.

In this section we discuss some results obtained by using the dictionaries learned via SAE training, but replacing the encoder with a different sparse approximation algorithm at inference time. This allows us to compare the dictionaries learned by different SAE training regimes independently of the quality of the encoder. It also allows us to examine the gap between the sparse reconstruction performed by the encoder against the baseline of a more powerful sparse approximation algorithm. As mentioned, for a fair comparison to the task the encoder is trained for, it is important to solve the sparse approximation problem of Eq. (12), rather than the more conventional formulation of Eq. (11), but most sparse approximation algorithms can be modified to solve this with relatively minor changes.

Solving Eq. (12) exactly is equivalent to integer linear programming, and is NP hard. The integer linear programs in question would be large, as our SAE decoders routinely have hundreds of thousands of features, and solving them to guaranteed optimality would likely be intractable. Instead, as is commonly done, we use iterative greedy algorithms to find an approximate solution. While the solution found by these sparse approximation algorithms is not guaranteed to be the global optimum, these are significantly more powerful than the SAE encoder, and we feel it is acceptable in practice to treat them as an upper bound on possible encoder performance.

For all results in this section, we use gradient pursuit, as described in Blumensath and Davies [6], as our inference time optimisation (ITO) algorithm. This algorithm is a variant of orthogonal matching pursuit [40] which solves the orgothonalisation of the residual to the span of chosen dictionary elements approximately at every step rather than exactly, but which only requires matrix multiplies rather than matrix solves and is easier to implement on accelerators as a result. It is possibly not crucial for performance that our optimisation algorithm be implementable on TPUs, but being able to avoid a host-device transfer when splicing this into the forward pass allowed us to re-use our existing evaluation pipeline with minimal changes.

When we use a sparse approximation algorithm at test time, we simply use the decoder of a trained SAE as a dictionary, ignoring the encoder. This allows us to sweep the target sparsity at test time without retraining the model, meaning that we can plot an entire Pareto frontier of loss recovered against sparsity for a single decoder, as in done in Fig. 7.

Fig. 6 compares the loss recovered when using ITO for a suite of SAEs decoders trained with both methods at three different test time L0 thresholds. This graph shows a somewhat surprising result; while Gated SAEs learn better decoders generally, and often achieve the best loss recovered using ITO close to their training sparsity, SAE decoders are often outperformed by decoders which achieved a higher test time L0; it's better to do ITO with a target L0 of 10 with an decoder with an achieved L0 of around 100 during training than one which was actually trained with this level of sparsity. For instance, the left hand panel in Fig. 6 shows that SAEs with a training L0 of 100 are better than those with an L0 of around 10 at almost every sparsity level in terms of ITO reconstruction. However, gated SAE dictionaries have a small but real advantage over standard SAEs in terms of loss recovered at most target sparsity levels, suggesting that part of the advantage of gated SAEs is that they learn better dictionaries as well as addressing issues with shrinkage. However, there are some subtleties here; for example, we find that baseline SAEs trained with a lower sparsity penalty (higher training L0) often outperform more sparse baseline SAEs according to this measure, and the best performing baseline SAE (L0 $\approx$ 99) is comparable to the best performing Gated SAE (L0 $\approx$ 20).

Fig. 7 compares the Pareto frontiers of a baseline model and a gated model to the Pareto frontier of an ITO sweep of the best performing dictionary of each. Note that, while the Pareto curve of the baseline dictionary is formed by several models as each encoder is specialised to a given sparsity

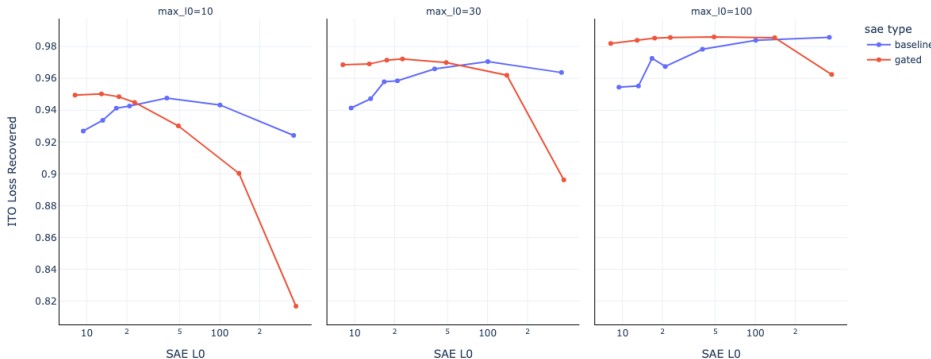

Figure 6: This figure compares the ITO performance of different decoders across a sweep for de-coders trained using a baseline SAE and the gated method, at three different test time target spar-sities. Gated SAEs trained at lower target sparsities consistently achieve better dictionaries by this measure. Interestingly, the best performing baseline dictionary by this measure often has a much higher test time sparsity than the target; for instance, at a test time sparsity of 30, the best baseline SAE was the one that had a test time sparsity of more like 100. This could be an artifact of the fact that the L0 measure is quite sensitive to noise, and standard SAE architectures tend to have a reasonable number of features with very low activation.

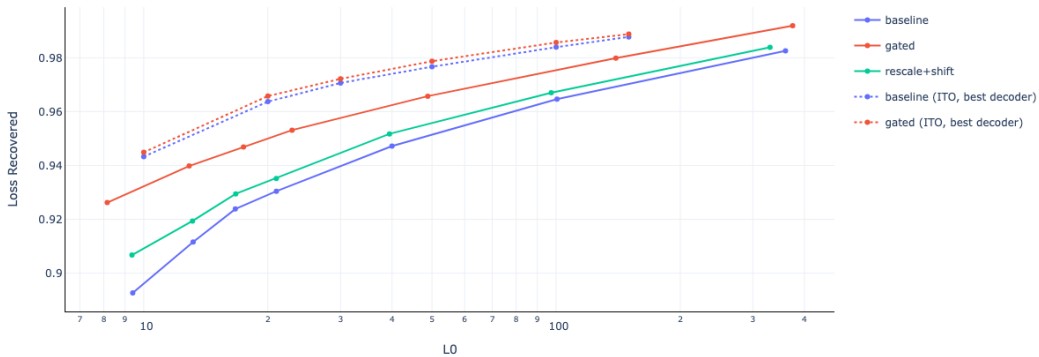

Figure 7: Pareto frontiers of a baseline SAE, a baseline SAE with learned rescale and shift (to account for shrinkage) and a gated SAE across different sparsity lambdas, compared to the ITO Pareto frontier of the best decoder of each type with ITO, varying the target sparsity. The best gated encoder is better than the best standard encoder by this measure, but the difference is marginal. As shown in the plot above, the best baseline encoder by the ITO measure had a much larger test time sparsity (around 100) than the best gated model (around 30). This figure suggests that the gap between SAE performance and 'optimal' performance, if we assume that ITO is close to the maximum possible reconstruction using the given encoder, is much smaller for the gated model.

level, as mentioned, ITO lets us plot a Pareto frontier by sweeping the target sparsity with a single dictionary; here we plot only the best performing dictionary from each model type to avoid cluttering the figure. This figure suggests that the performance gap between the encoder and using ITO is smaller for the gated model. Interestingly, this cannot solely be explained by addressing shrinkage, as we demonstrate by experimenting with a baseline model which learns a rescale and shift with a frozen encoder and decoder directions.

Table 1: Cross-entropy losses for the original language model and after zero-ablating specified sub-layers of Pythia-2.8B and Gemma-7B.

| Model | Layer | Original CE Loss | Zero Ablation CE Loss | | |
| --- | --- | --- | --- | --- | --- |
| | | | MLP | Attention | Residual |
| Gemma-7B (1024 length context) | 6 | 2.5426 | 2.7764 | 2.7295 | 16.1549 |
| | 13 | 2.5426 | 2.5878 | 2.5566 | 30.3588 |
| | 20 | 2.5426 | 2.6881 | 2.5726 | 19.5891 |
| | 27 | 2.5426 | 26.1114 | 3.0819 | 12.4534 |
| Pythia-2.8B (2048 length context) | 4 | 1.9699 | 2.0460 | 2.0361 | 13.0434 |
| | 12 | 1.9699 | 2.0167 | 2.0131 | 10.6558 |
| | 16 | 1.9699 | 2.0098 | 2.0046 | 11.6820 |
| | 20 | 1.9699 | 2.1022 | 2.0269 | 10.4578 |
| | 28 | 1.9699 | 2.0145 | 1.9760 | 27.8663 |

# E    More loss recovered / L0 Pareto frontiers

In Fig. 8 we show that Gated SAEs outperform baseline SAEs. In Fig. 9 we show that Gated SAEs ourperform baseline SAEs at all but one MLP output or residual stream site that we tested on.

In Fig. 9 at the attention output pre-linear site at layer 27, loss recovered is bigger than 1.0. On investigation, we found that the dataset used to train the SAE was not identical to Gemma's pretraining dataset, and at this site it was possible to mean ablate this quantity and decrease loss – explaining why SAE reconstructions had lower loss than the original model.

Table 1 provides cross-entropy losses for the Gemma-7B and Pythia-2.8B, both before and after zero-ablating specific sub-layers of these models, to help provide further context for interpreting the loss recovered results presented in this paper; 100% loss recovered corresponds to the SAE-spliced language model attaining a loss matching the original language model, whereas 0% loss recovered corresponds to the SAE-spliced language model attaining a loss matching the language model with the corresponding sub-layer zero-ablated.

# F    Further shrinkage plots

In Fig. 10, we show that Gated SAEs resolve shrinkage, as measured by relative reconstruction bias (Appendix C), in Pythia-2.8B.

# G    Training and evaluation: hyperparameters and other details

## G.1    Training

### G.1.1    General training details

Other details of SAE training are:

- **SAE Widths**. Our SAEs have width $2^{17}$ for most baseline SAEs, $3 \times 2^{16}$ for Gated SAEs, except for the (Pythia-2.8B, Residual Stream) sites we used $2^{15}$ for baseline and $3 \times 2^{14}$ for Gated since early runs at these sites had lots of learned feature death.
- **Training data**. We use activations from hundreds of millions to billions of activations from LM forward passes as input data to the SAE. Following Nanda [29], we use a shuffled buffer of these activations, so that optimization steps don't use data from highly correlated activations.[14]
- **Resampling**. We used *resampling*, a technique which at a high-level reinitializes features that activate extremely rarely on SAE inputs periodically throughout training. We mostly

---

[14]In contrast to earlier findings [9], we found that when using Pythia-2.8B's activations from sequences of length 2048, rather than GELU-1L's activations from sequences of length 128, it was important to shuffle the $10^6$ length activation buffer used to train our SAEs.

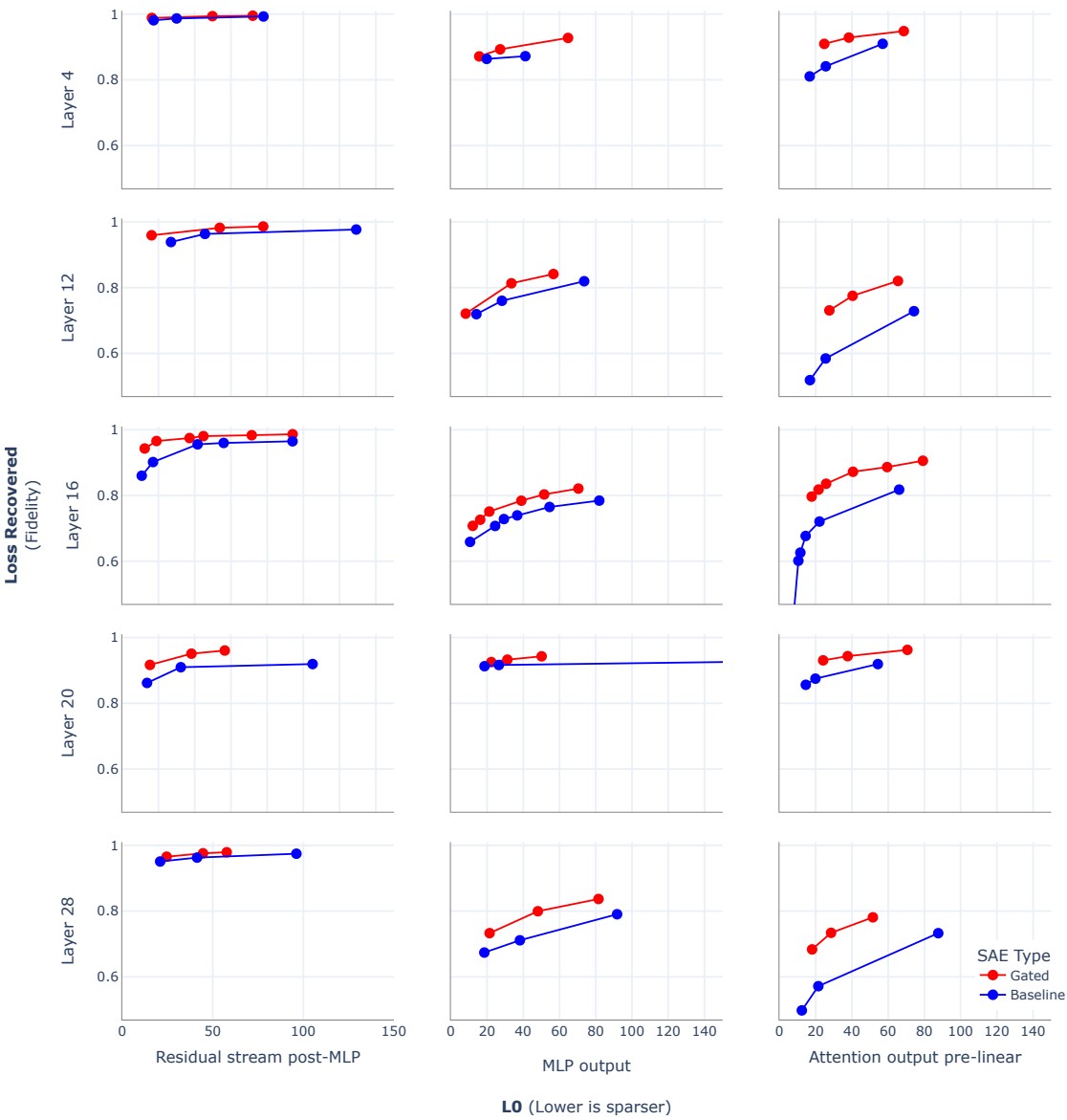

Figure 8: Gated SAEs throughout Pythia-2.8B. At all sites we tested, Gated SAEs are a Pareto improvement. In every plot, the SAE with maximal loss recovered was a Gated SAE.

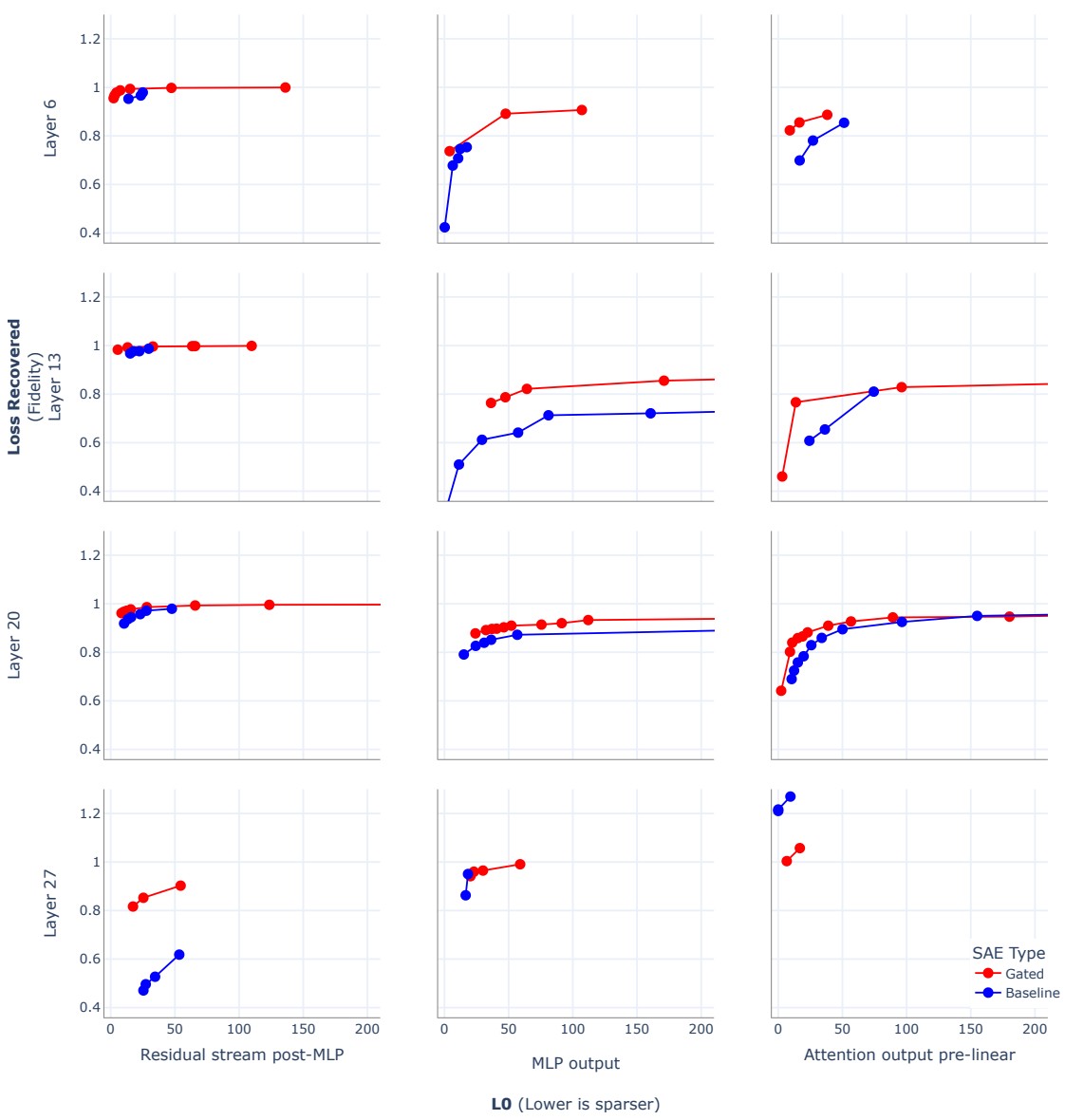

Figure 9: Gated and Normal Pareto-Optimal SAEs for Gemma-7B – see Appendix E for a discussion of the anomalies (such as the Layer 27 attention output SAEs).

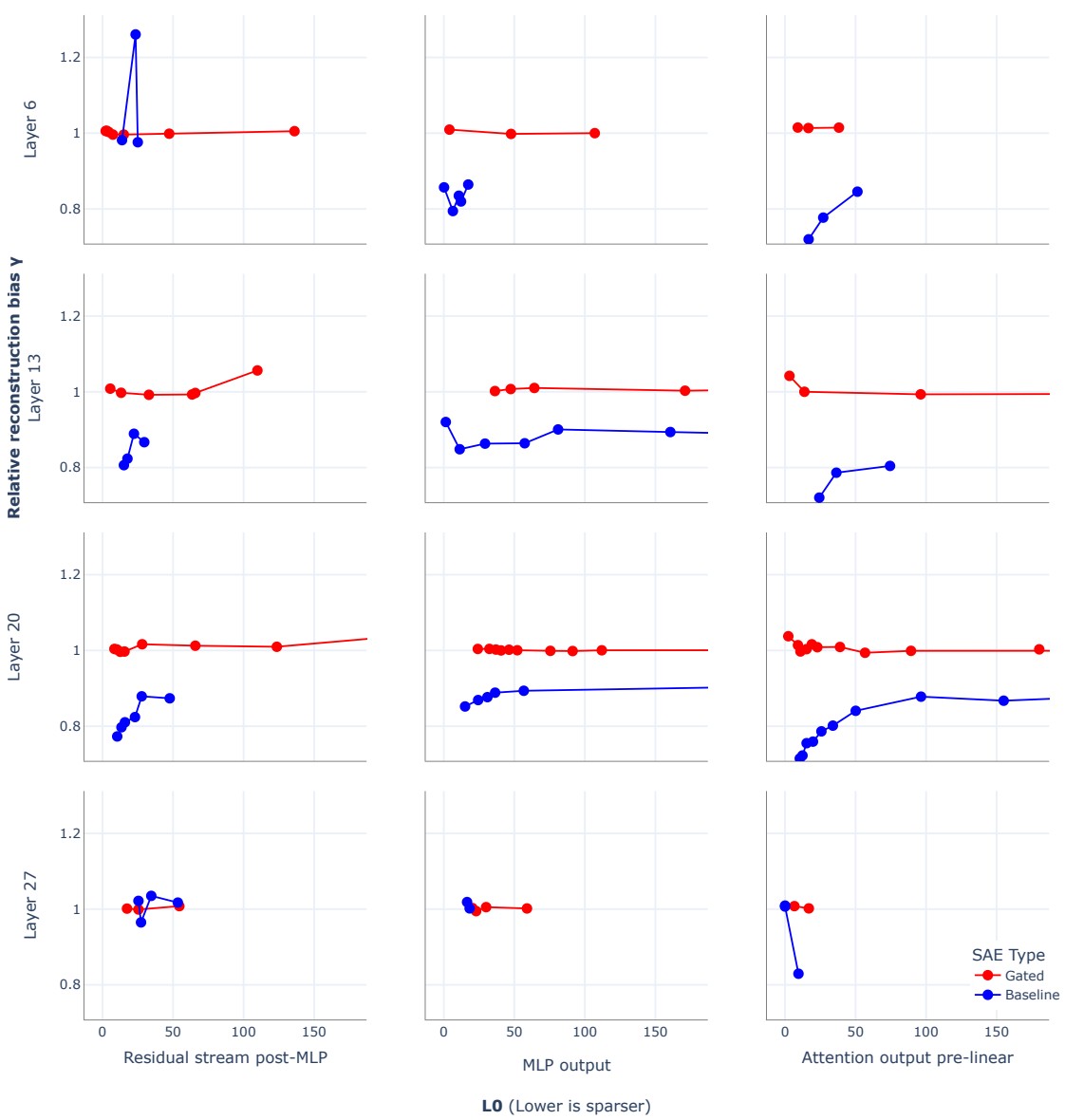

Figure 10: Gated SAEs address the problem of shrinkage in Pythia-2.8B.

follow the approach described in the 'Neuron Resampling' appendix of Bricken et al. [8], except we reapply learning rate warm-up after each resampling event, reducing learning rate to 0.1x the ordinary value, and, increasing it with a cosine schedule back to the ordinary value over the next 1000 training steps.

- **Optimizer hyperparameters**. We use the Adam optimizer with $\beta_2 = 0.999$ and $\beta_1 = 0.0$, following Templeton et al. [47], as we also find this to be a slight improvement to training. We use a learning rate warm-up. See Appendix G.1.2 for learning rates of different experiment.
- **Decoder weight norm constraints**. Templeton et al. [47] suggest constraining columns to have *at most* unit norm (instead of exactly unit norm), which can help distinguish between productive and unproductive feature directions (although it should have no systematic impact on performance). However, we follow the original approach of constraining columns to have exact unit norms in this work for the sake of simplicity.
- **Compute resources**. Individual SAEs were each trained on TPU-v3 slices with a 2x2 topology [20]. The same chips were used to generate LM activations on-the-fly, train SAE parameters and evaluate SAEs during training, using up to 8-way model parallelism. With this setup, the time to train a SAE varies by SAE width, LM residual stream dimension, sequence length, layer and site.[15] We also used a negligible amount of compute on resampling (Appendix G), evaluation (e.g. Figure 1) and interpretability experiments (Section 4.2). Training wall clock time ranges from around 7 hours to train on GELU-1L MLP activations to around 47 hours to train on Gemma-7B sites at layer 27. We estimate that we used twice as much compute as used in the paper on preliminary experiments.

### G.1.2 Experiment-specific training details

- We use learning rate 0.0003 for all Gated SAE experiments, and the GELU-1L baseline experiment. We swept for optimal baseline learning rates for the GELU-1L baseline to generate this value. For the Pythia-2.8B and Gemma-7B baseline SAE experiments, we divided the L2 loss by $\mathbb{E}||x||_2$, motivated by better hyperparameter transfer, and so changed learning rate to 0.001 and 0.00075. We didn't see noticeable difference in the Pareto frontier and so did not sweep this hyperparameter further.
- We generate activations from sequences of length 128 for GELU-1L, 2048 for Pythia-2.8B and 1024 for Gemma-7B.
- We use a batch size of 4096 for all runs. We use 300,000 training steps for GELU-1L and Gemma-7B runs, and 400,000 steps for Pythia-2.8B runs.

### G.1.3 Lessons learned scaling SAEs

- **Learned feature death is unpredictable**. In Fig. 11 there are few patterns that can be gleaned from staring at which runs have high numbers of dead learned features (called dead neurons in Bricken et al. [8]).
- **Resampling makes hyperparameter sweeps difficult**. We found that resampling caused L0 and loss recovered to increase, similar to Conmy [9].
- **Training appears to converge earlier than expected**. We found that we did not need 20B tokens as in Bricken et al. [8], as generally resampling had stopped causing gains and loss curves plateaued after just over one billion tokens.

### G.2 Evaluation

We evaluated the models on over a million held-out tokens.

# H Equivalence between gated encoder with tied weights and linear encoder with non-standard activation function

In this section we show under the weight sharing scheme defined in Eq. (7), a gated encoder as defined in Eq. (5) is equivalent to a linear layer with a non-standard (and parameterized) activation function.

Without loss of generality, consider the case of a single latent feature ($M = 1$) and set the pre-encoder bias to zero. In this case, the gated encoder is defined as

$$\tilde{f}(\mathbf{x}) := \mathbb{1}_{\mathbf{w}_{\text{gate}} \cdot \mathbf{x} + b_{\text{gate}} > \mathbf{0}} \, \text{ReLU} \left( \mathbf{w}_{\text{mag}} \cdot \mathbf{x} + b_{\text{mag}} \right) \tag{13}$$

and the weight sharing scheme becomes

$$\mathbf{w}_{\text{mag}} := \rho_{\text{mag}} \mathbf{w}_{\text{gate}} \tag{14}$$

with a non-negative parameter $\rho_{\text{mag}} \equiv \exp(\mathbf{r}_{\text{mag}})$.

Substituting Eq. (14) into Eq. (13) and re-arranging, we can re-express $\tilde{f}(\mathbf{x})$ as a single linear layer

$$\tilde{f}(\mathbf{x}) := \sigma_{b_{\text{mag}} - \rho_{\text{mag}} b_{\text{gate}}} \left( \mathbf{w}_{\text{mag}} \cdot \mathbf{x} + b_{\text{mag}} \right) \tag{15}$$

with the parameterized activation function

$$\sigma_\theta(z) := \mathbb{1}_{z > \theta} \, \text{ReLU} \left( z \right). \tag{16}$$

called JumpReLU in a different context [17]. Fig. 12 illustrates the shape of this activation function.

# I Further analysis of the human interpretability study

We perform some further analysis on the data from Section 4.2, to understand the impact of different sites, layers, and raters.

## I.1 Sites

We first pose the question of whether there's evidence that the sites had different interpretability outcomes. A Friedman test across sites shows significant differences (at $p = 0.047$) between the Gated-vs-Baseline differences, though not ($p = 0.92$) between the raw labels.

Breaking down by site and repeating the Wilcoxon-Pratt one-sided tests and computing confidence intervals, we find the result on MLP outputs is strongest, with mean 0.40, significance $p = 0.003$, and CI [0.18, 0.63]; this is as compared with the attention outputs ($p = 0.47$, mean .05, CI [-0.16, 0.26]) and final residual ($p = 0.59$, mean -0.07, CI [-0.28, 0.12]) SAEs.

## I.2 Layers

Next we test whether different layers had different outcomes. We do this separately for the 2 models, since the layers aren't directly comparable. We run 2 tests in each setting: Page's trend test (which tests for a monotone trend across layers) and the Friedman test (which tests for any difference, without any expectation of a monotone trend).

Results are presented in Table 2; they suggest there are some significant nonmonotone differences between layers. To elucidate this, we present 90% BCa bootstrap confidence intervals of the mean raw label (where 'No'=0, 'Maybe'=1, 'Yes'=2) and the Gated-vs-Baseline difference, per layer, in Fig. 15 and Fig. 16, respectively.

## I.3 Raters

In Table 3 we present test results weakly suggesting that the raters differed in their judgments. This underscores that there's still a significant subjective component to this interpretability labeling. (Notably, different raters saw different proportions of Pythia vs Gemma features, so aggregating across the models is partially confounded by that.)

---

[15]The FLOPs required to compute LM activations increase with layer; SAEs trained on MLP activations have a higher parameter count than those trained on MLP outputs, attention outputs or the residual stream.

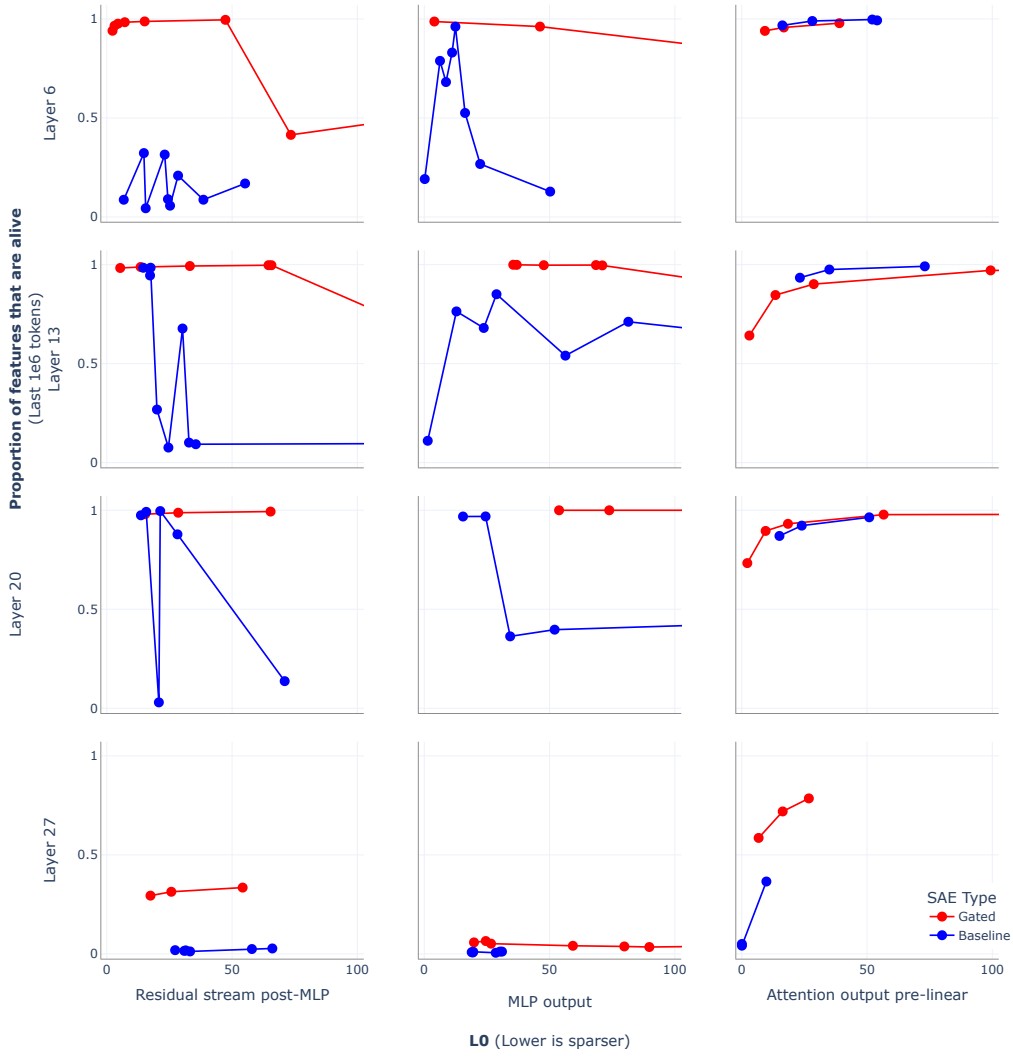

Figure 11: Feature death in Gemma-7B.

| $p$-values | Raw label | Delta from Baseline to Gated |
|---|---|---|
| Pythia-2.8B (Page's trend test) | 0.50 | 0.13 |
| Pythia-2.8B (Friedman test) | 0.57 | 0.05 |
| Gemma-7B (Page's trend test) | 0.037 | 0.31 |
| Gemma-7B (Friedman test) | 0.003 | 0.64 |

Table 2: Layer significance tests

| $p$-values | Raw label | Delta from Baseline to Gated |
|---|---|---|
| Across models (Kruskal-Wallis H-test) | 0.01 | 0.71 |
| Pythia-2.8B (Friedman test) | 0.13 | 0.05 |
| Gemma-7B (Friedman test) | 0.03 | 0.76 |

Table 3: Rater significance tests

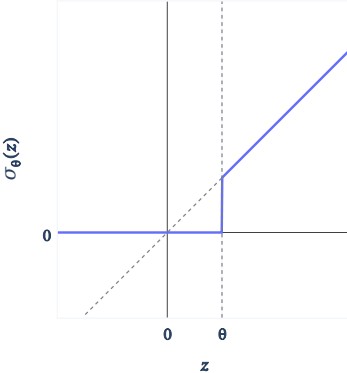

Figure 12: After applying the weight sharing scheme of Eq. (7), a gated encoder becomes equivalent to a single layer linear encoder with a JumpReLU (Erichson et al. [17], previously named TRec by Konda et al. [23]) activation function $\sigma_\theta$, illustrated above.

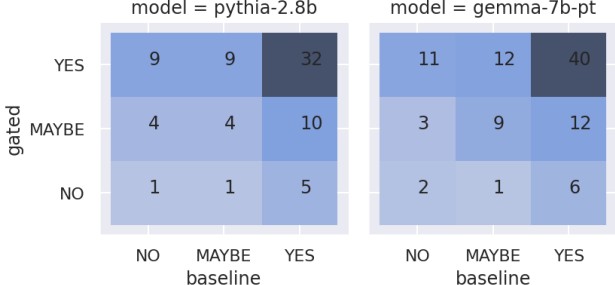

Figure 13: Contingency table showing Gated vs Baseline interpretability labels from our paired study results, for Pythia-2.8B and Gemma-7B.

**ACTIVATIONS**
**DENSITY = 0.122%**

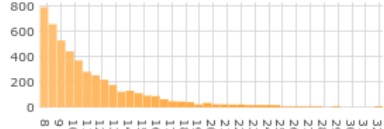

| NEGATIVE LOGITS | | POSITIVE LOGITS | |
|---|---|---|---|
| EOF | -0.08 | above | +0.53 |
| cheat | -0.08 | above | +0.46 |
| tomat | -0.08 | aforementioned | +0.38 |
| öt | -0.07 | foregoing | +0.37 |
| getElement | -0.07 | mentioned | +0.36 |
| invoke | -0.07 | Above | +0.35 |
| strugg | -0.07 | Above | +0.33 |
| lie | -0.07 | described | +0.31 |
| compute | -0.07 | mentioned | +0.31 |
| anean | -0.07 | described | +0.26 |

**(FEATURES * UNEMBED) HISTOGRAM**

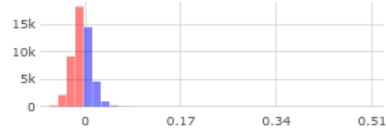

**Activations in range 29.0299 to 32.2555**

this tameness to its extreme voracity. Be this as it may ; when a flock of these birds are feeding in a pine-tree in the manner described , they
nutrition labels ; the Korea Food Research Institute ; and revised data from the Rural Development Administration's research analysis. For foods that were not found in any database mentioned above, we

**Activations in range 25.8044 to 29.0299**

by Manning Oil & Gas will pass to Permina upon entry into Indonesia, with Manning 's out of pocket costs to be recovered by the forty percent formula ."\n\n
velocity distribution in the depth direction .\n Patent Document 1 focuses on the rising and sinking of the ocean current power generation device depending on the current velocity as described above, and
terminal drops down to the same lower level. When both terminals are transmitting the same flag sequence this step completes .\n Arising from the set of procedures described above that are
other nation. The Government of Pakistan will not, without the prior agreement of the Government of the United •States, devote such assistance to purposes other than those for which it
ed foreign exchange, import restrictions, and importers inexperienced in assessing technology or in export marketing. In contrast, firms whose characteristics are the opposite of those enumerated above are
vessel is a ship not operating on a regular route or schedule ; that is, tramp steamers do not have the established schedules of the other two types of carriers.
, a trench 45 corresponding to the opening 43 is formed in the silicon base material 40 (FIG. 4B).\n However, during the dry etching described above, SiO
and I hereby promise to ask Mr. Long to excuse us if we ignore him more in the future .\n\n Without any one of the friends and family listed here —
de la Pena and A. Cetto, E. Santos, O. Theimer, and G. Goedecke ; see the reviews mentioned above. More
eding strategy .[]{data-label="fig::initialisation 5"}](initialisation 5.pdf){width="50.00000%"} \n\n The hedging described above is carried

**Activations in range 22.5788 to 25.8044**

7 entitled "The Introduction of Low Erucic Acid Rapeseed Varieties Into Canadian Production" by J. K. Daun from the previously identified Academic Press
in a case where, for example, a coronary artery (blood vessel) present in the vicinity of the heart is observed with use of the manual tracking method by the operator
, and total medical expense management .\n\n We are fortunate to live in an area with some of the best hospitals in the nation. The approaches I have outlined for improved
10 kIU/ml, TUDCA: 1, 5, 15 mg/ml ; pH 5.0) were obtained using the method described and by
of two subunits whose binding is contingent on phosphorylation by separately activated kinases .\n 3. Effects of Errors \n Errors in the signal transduction and regulation pathways described above can cause
will feel its force. But can their irresistible skepticism concerning poetry translation and paraphrase be reconciled with the obvious logic of the simple refutation ?\n\n
verdict judgment of acquittal as to yet another\n count. In the end, Kim was convicted only on counts six and\n seven, covering the incidents described above. \n\n
propose a new algorithmic approach, exhibited in Fig. \[fig:teaser\], that improves upon the SLIC approach, motivated by the drawbacks discussed above. We
a collection of functions that sometimes call functions in other packages (where some bit of work has already been coded-up). Is this scenario 1 or 2 above ?\n
substitutions can change the aesthetic, emotive or imagistic quality of a poem, how could any of them change meaning ?\n\n Despite the simple refutation, the he

Figure 14: An extract of a feature visualization dashboard used to rate features in the interpretability study described in Section 4.2. The left-hand pane provides aggregate information, including a feature histogram and the tokens most promoted and demoted by the feature being rated. The rest of the dashboard displays samples of text on which the feature activates to various degrees. Holding the mouse over a text token reveals a hover showing the exact activation level at that token. Although not shown here, the full dashboard provides examples across the full range of activations, down to examples on which the feature fails to activate. This particular feature, taken from a layer 20 Gemma-7B residual stream Gated SAE seems to promote completions like "above", "aforementioned", "mentioned above" etc. in contexts where such a completion would be likely.

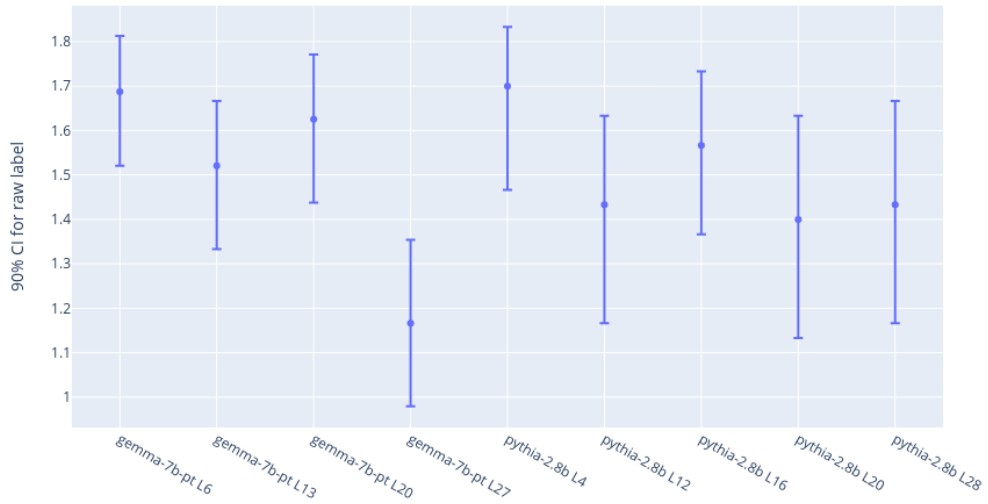

Figure 15: Per-layer 90% confidence intervals for the mean interpretability label

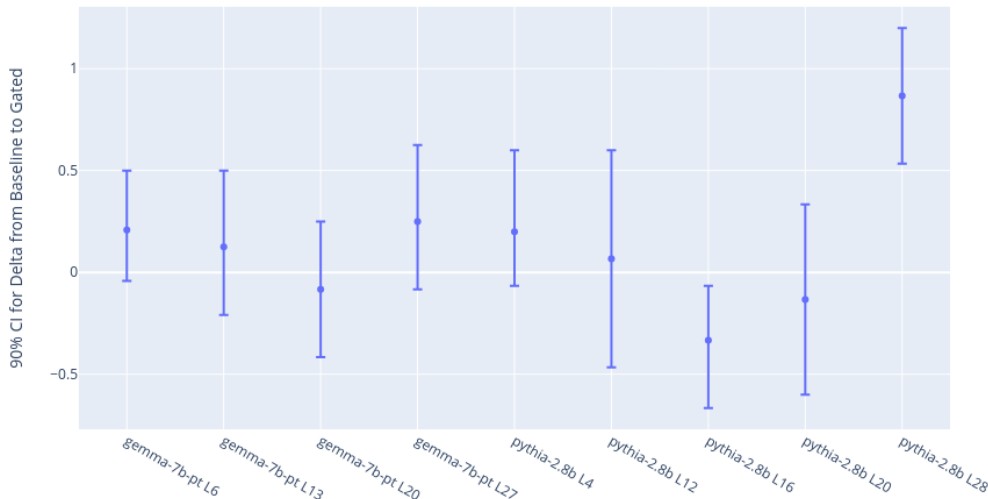

Figure 16: Per-layer 90% confidence intervals for the Gated-vs-Baseline label difference

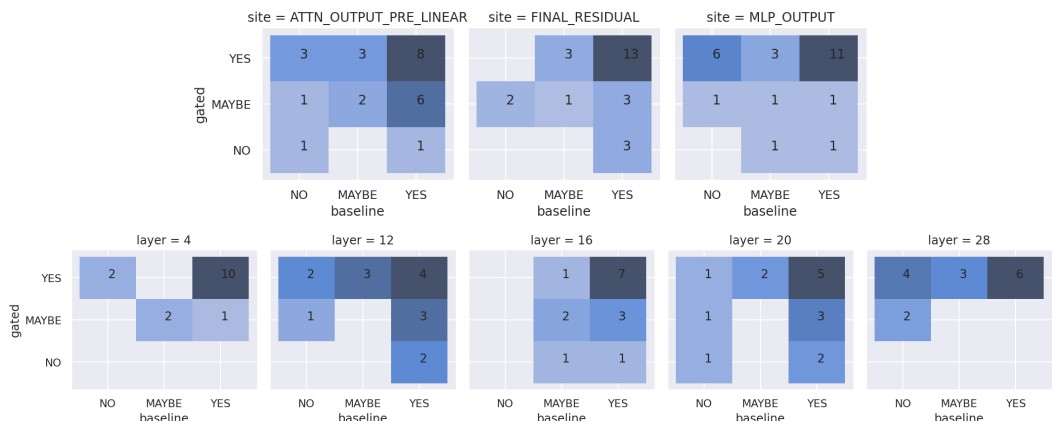

Figure 17: Contingency tables for the paired (gated vs baseline) interpretability labels, for Pythia-2.8B

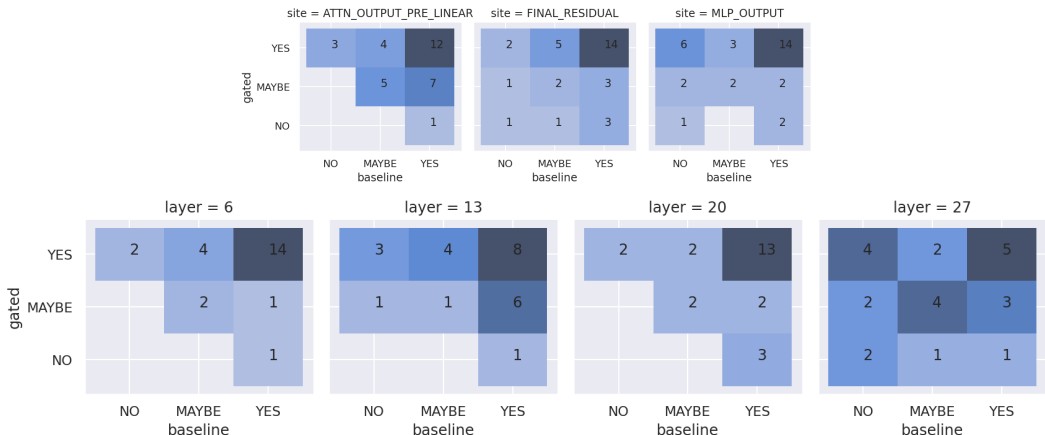

Figure 18: Contingency tables for the paired (gated vs baseline) interpretability labels, for Gemma-7B

# J  Pseudo-code for Gated SAEs and the Gated SAE loss function

```python
def gated_sae(x, W_gate, b_gate, W_mag, b_mag, W_dec, b_dec):
  # Apply pre-encoder bias
  x_center = x - b_dec

  # Gating encoder (estimates which features are active)
  active_features = ((x_center @ W_gate + b_gate) > 0)

  # Magnitudes encoder (estimates active features' magnitudes)
  feature_magnitudes = relu(x_center @ W_mag + b_mag)

  # Multiply both before decoding
  return (active_features * feature_magnitudes) @ W_dec + b_dec
```

Figure 19: Pseudo-code for the Gated SAE forward pass.

```python
def loss(x, W_gate, b_gate, W_mag, b_mag, W_dec, b_dec):
  gated_sae_loss = 0.0

  # We'll use the reconstruction from the baseline forward pass to train
  # the magnitudes encoder and decoder. Note we don't apply any sparsity
  # penalty here. Also, no gradient will propagate back to W_gate or b_gate
  # due to binarising the gated activations to zero or one.
  reconstruction = gated_sae(x, W_gate, b_gate, W_mag, b_mag, W_dec, b_dec)
  gated_sae_loss += sum((reconstruction - x)**2, axis=-1)

  # We apply a L1 penalty on the gated encoder activations (pre-binarising,
  # post-ReLU) to incentivise them to be sparse
  x_center = x - b_dec
  via_gate_feature_magnitudes = relu(x_center @ W_gate + b_gate)
  gated_sae_loss += l1_coef * sum(via_gate_feature_magnitudes, axis=-1)

  # Currently the gated encoder only has gradient signal to be sparse, and
  # not to reconstruct well, so we also do a "via gate" reconstruction, to
  # give it an appropriate gradient signal. We stop the gradients to the
  # decoder parameters in this forward pass, as we don't want these to be
  # influenced by this auxiliary task.
  via_gate_reconstruction = (
    via_gate_feature_magnitudes @ stop_gradient(W_dec)
    + stop_gradient(b_dec)
  )
  gated_sae_loss += sum((via_gate_reconstruction - x)**2, axis=-1)

  return gated_sae_loss
```

Figure 20: Pseudo-code for the Gated SAE loss function. Note that this pseudo-code is written for expositional clarity. In practice, taking into account parameter tying, it would be more efficient to rearrange the computation to avoid unnecessarily duplicated operations.

# K  Tables of Results

We evaluated the models on over a million held-out tokens. Tables 4-11 show summary stats from training runs on the Pareto frontier.

| Site | Layer | Sparsity $\lambda$ | LR | L0 | % CE Recovered | Clean CE Loss | SAE CE Loss | 0 Abl. CE Loss | Width | % Alive Features | Shrinkage $\gamma$ |
|---|---|---|---|---|---|---|---|---|---|---|---|
| *Resid* | *6* | *3e-05* | *0.001* | *18.1* | *95.28%* | *2.5426* | *3.1847* | *16.1549* | *196608* | *16.8%* | *0.982* |
| Resid | 6 | 2e-05 | 0.001 | 10.5 | 85.3% | 2.5426 | 4.5433 | 16.1549 | 196608 | 5.72% | 1.136 |
| Resid | 6 | 1e-05 | 0.001 | 19.0 | 91.24% | 2.5426 | 3.7349 | 16.1549 | 196608 | 5.11% | 1.606 |
| *Resid* | *6* | *2e-05* | *0.00075* | *29.8* | *96.65%* | *2.5426* | *2.9989* | *16.1549* | *196608* | *13.67%* | *1.261* |
| *Resid* | *6* | *3e-05* | *0.00075* | *25.4* | *97.9%* | *2.5426* | *2.8279* | *16.1549* | *196608* | *38.86%* | *0.976* |
| Resid | 6 | 8e-06 | 0.00075 | 29.8 | 91.28% | 2.5426 | 3.7301 | 16.1549 | 196608 | 9.88% | 1.105 |
| Resid | 6 | 1e-05 | 0.00075 | 57.3 | 97.36% | 2.5426 | 2.9023 | 16.1549 | 196608 | 11.78% | 1.03 |
| Resid | 6 | 4e-06 | 0.00075 | 69.2 | 95.98% | 2.5426 | 3.0892 | 16.1549 | 196608 | 13.54% | 1.239 |
| Resid | 6 | 6e-06 | 0.00075 | 40.0 | 95.49% | 2.5426 | 3.1562 | 16.1549 | 196608 | 24.34% | 1.159 |
| *Resid* | *13* | *9e-05* | *0.00075* | *14.3* | *96.77%* | *2.5426* | *3.4423* | *30.3588* | *196608* | *98.38%* | *0.806* |
| *Resid* | *13* | *8e-05* | *0.00075* | *17.5* | *97.66%* | *2.5426* | *3.1947* | *30.3588* | *196608* | *98.7%* | *0.824* |
| Resid | 13 | 8e-05 | 0.001 | 18.0 | 97.63% | 2.5426 | 3.2021 | 30.3588 | 196608 | 95.35% | 0.838 |
| *Resid* | *13* | *5e-05* | *0.00075* | *22.2* | *97.69%* | *2.5426* | *3.1849* | *30.3588* | *196608* | *25.78%* | *0.889* |
| Resid | 13 | 3e-05 | 0.00075 | 29.0 | 97.64% | 2.5426 | 3.1986 | 30.3588 | 196608 | 8.55% | 0.903 |
| *Resid* | *13* | *5e-05* | *0.001* | *29.5* | *98.71%* | *2.5426* | *2.9005* | *30.3588* | *196608* | *65.17%* | *0.867* |
| Resid | 13 | 3e-05 | 0.001 | 39.2 | 98.26% | 2.5426 | 3.026 | 30.3588 | 196608 | 26.33% | 0.936 |
| Resid | 13 | 2e-05 | 0.00075 | 56.6 | 98.49% | 2.5426 | 2.9615 | 30.3588 | 196608 | 16.19% | 0.976 |
| Resid | 13 | 1e-05 | 0.00075 | 101.3 | 97.83% | 2.5426 | 3.1459 | 30.3588 | 196608 | 4.55% | 1.018 |
| *Resid* | *20* | *0.00012* | *0.00075* | *10.4* | *91.87%* | *2.5426* | *3.9277* | *19.5891* | *196608* | *92.51%* | *0.773* |
| *Resid* | *20* | *0.0001* | *0.00075* | *13.8* | *93.68%* | *2.5426* | *3.6204* | *19.5891* | *196608* | *97.46%* | *0.797* |
| *Resid* | *20* | *9e-05* | *0.00075* | *16.0* | *94.48%* | *2.5426* | *3.4835* | *19.5891* | *196608* | *99.2%* | *0.81* |
| Resid | 20 | 3e-05 | 0.001 | 25.2 | 90.71% | 2.5426 | 4.1258 | 19.5891 | 196608 | 3.11% | 0.951 |
| *Resid* | *20* | *7e-05* | *0.001* | *21.3* | *95.73%* | *2.5426* | *3.27* | *19.5891* | *196608* | *99.62%* | *0.824* |
| *Resid* | *20* | *5e-05* | *0.001* | *27.8* | *97.15%* | *2.5426* | *3.0281* | *19.5891* | *196608* | *88.4%* | *0.879* |
| Resid | 20 | 3e-05 | 0.00075 | 39.1 | 96.43% | 2.5426 | 3.1518 | 19.5891 | 196608 | 35.64% | 1.019 |
| *Resid* | *20* | *4e-05* | *0.00075* | *46.4* | *97.95%* | *2.5426* | *2.8922* | *19.5891* | *196608* | *99.9%* | *0.874* |
| Resid | 20 | 2e-05 | 0.00075 | 49.4 | 95.26% | 2.5426 | 3.3505 | 19.5891 | 196608 | 8.61% | 0.983 |
| Resid | 20 | 1.5e-05 | 0.00075 | 50.3 | 95.99% | 2.5426 | 3.2268 | 19.5891 | 196608 | 9.46% | 2.179 |
| Resid | 20 | 1e-05 | 0.00075 | 124.8 | 97.69% | 2.5426 | 2.9367 | 19.5891 | 196608 | 12.3% | 0.997 |
| *Resid* | *27* | *1e-05* | *0.001* | *27.6* | *47.08%* | *2.5426* | *7.7878* | *12.4534* | *196608* | *1.68%* | *1.022* |
| *Resid* | *27* | *8e-06* | *0.001* | *30.5* | *49.63%* | *2.5426* | *7.5345* | *12.4534* | *196608* | *1.12%* | *0.965* |
| Resid | 27 | 1.2e-05 | 0.00075 | 36.2 | 39.49% | 2.5426 | 8.5398 | 12.4534 | 196608 | 2.02% | 1.564 |
| *Resid* | *27* | *6e-06* | *0.001* | *39.1* | *52.72%* | *2.5426* | *7.228* | *12.4534* | *196608* | *1.39%* | *1.035* |
| *Resid* | *27* | *4e-06* | *0.00075* | *63.4* | *61.84%* | *2.5426* | *6.3246* | *12.4534* | *196608* | *3.03%* | *1.017* |
| Resid | 27 | 2e-06 | 0.00075 | 88.2 | 58.45% | 2.5426 | 6.6609 | 12.4534 | 196608 | 2.22% | 1.163 |
| *MLP* | *6* | *0.0004* | *0.001* | *0.2* | *42.33%* | *2.5426* | *2.6774* | *2.7764* | *196608* | *19.17%* | *0.857* |
| *MLP* | *6* | *0.0001* | *0.001* | *6.3* | *67.78%* | *2.5426* | *2.6179* | *2.7764* | *196608* | *82.35%* | *0.794* |
| MLP | 6 | 0.0001 | 0.00075 | 7.6 | 59.55% | 2.5426 | 2.6371 | 2.7764 | 196608 | 69.88% | 1.189 |
| *MLP* | *6* | *7e-05* | *0.001* | *10.6* | *70.77%* | *2.5426* | *2.6109* | *2.7764* | *196608* | *75.8%* | *0.835* |
| MLP | 6 | 3e-05 | 0.00075 | 15.3 | 64.49% | 2.5426 | 2.6256 | 2.7764 | 196608 | 15.36% | 1.001 |
| *MLP* | *6* | *7e-05* | *0.00075* | *12.0* | *74.63%* | *2.5426* | *2.6019* | *2.7764* | *196608* | *94.97%* | *0.82* |
| MLP | 6 | 1.5e-05 | 0.00075 | 14.9 | 47.57% | 2.5426 | 2.6651 | 2.7764 | 196608 | 3.03% | 1.0 |
| *MLP* | *6* | *5e-05* | *0.00075* | *17.1* | *75.36%* | *2.5426* | *2.6002* | *2.7764* | *196608* | *68.12%* | *0.864* |
| *MLP* | *13* | *8e-05* | *0.00075* | *1.4* | *32.78%* | *2.5426* | *2.573* | *2.5878* | *196608* | *10.16%* | *0.92* |
| *MLP* | *13* | *8e-05* | *0.001* | *11.3* | *50.99%* | *2.5426* | *2.5647* | *2.5878* | *196608* | *73.07%* | *0.848* |
| MLP | 13 | 5e-05 | 0.001 | 22.6 | 47.32% | 2.5426 | 2.5664 | 2.5878 | 196608 | 66.09% | 0.882 |
| *MLP* | *13* | *5e-05* | *0.00075* | *29.4* | *61.19%* | *2.5426* | *2.5601* | *2.5878* | *196608* | *84.51%* | *0.863* |
| *MLP* | *13* | *3e-05* | *0.001* | *44.8* | *64.14%* | *2.5426* | *2.5588* | *2.5878* | *196608* | *56.91%* | *0.864* |
| *MLP* | *13* | *3e-05* | *0.00075* | *80.8* | *71.28%* | *2.5426* | *2.5556* | *2.5878* | *196608* | *73.31%* | *0.901* |
| *MLP* | *13* | *2e-05* | *0.00075* | *160.7* | *72.08%* | *2.5426* | *2.5552* | *2.5878* | *196608* | *56.12%* | *0.894* |
| *MLP* | *13* | *1e-05* | *0.00075* | *610.0* | *77.67%* | *2.5426* | *2.5527* | *2.5878* | *196608* | *44.39%* | *0.858* |
| *MLP* | *20* | *7e-05* | *0.001* | *15.8* | *79.11%* | *2.5426* | *2.573* | *2.6881* | *196608* | *96.84%* | *0.852* |
| *MLP* | *20* | *5e-05* | *0.001* | *24.5* | *82.67%* | *2.5426* | *2.5678* | *2.6881* | *196608* | *96.93%* | *0.869* |

Table 4: Gemma-7B Baseline SAEs (1024 sequence length). Italic are Pareto optimal SAEs.

| Site | Layer | Sparsity $\lambda$ | LR | L0 | % CE Recovered | Clean CE Loss | SAE CE Loss | 0 Abl. CE Loss | Width | % Alive Features | Shrinkage $\gamma$ |
|---|---|---|---|---|---|---|---|---|---|---|---|
| MLP | 20 | 5e-05 | 0.00075 | 26.0 | 82.36% | 2.5426 | 2.5682 | 2.6881 | 196608 | 97.96% | 0.865 |
| *MLP* | *20* | *4.5e-05* | *0.00075* | *31.4* | *83.94%* | *2.5426* | *2.5659* | *2.6881* | *196608* | *99.24%* | *0.877* |
| MLP | 20 | 3e-05 | 0.001 | 39.5 | 83.12% | 2.5426 | 2.5671 | 2.6881 | 196608 | 46.33% | 0.924 |
| *MLP* | *20* | *4e-05* | *0.00075* | *38.3* | *85.18%* | *2.5426* | *2.5641* | *2.6881* | *196608* | *95.73%* | *0.889* |
| MLP | 20 | 3.5e-05 | 0.00075 | 43.2 | 84.11% | 2.5426 | 2.5657 | 2.6881 | 196608 | 94.62% | 0.874 |
| *MLP* | *20* | *3e-05* | *0.00075* | *56.8* | *87.23%* | *2.5426* | *2.5612* | *2.6881* | *196608* | *96.88%* | *0.894* |
| MLP | 20 | 2e-05 | 0.00075 | 68.1 | 84.18% | 2.5426 | 2.5656 | 2.6881 | 196608 | 53.42% | 0.898 |
| MLP | 20 | 2e-05 | 0.00075 | 75.6 | 85.63% | 2.5426 | 2.5635 | 2.6881 | 196608 | 66.29% | 0.899 |
| MLP | 20 | 1.5e-05 | 0.00075 | 104.6 | 85.71% | 2.5426 | 2.5634 | 2.6881 | 196608 | 41.7% | 0.965 |
| *MLP* | *20* | *1e-05* | *0.00075* | *321.1* | *90.3%* | *2.5426* | *2.5567* | *2.6881* | *196608* | *56.83%* | *0.911* |
| *MLP* | *27* | *1.2e-05* | *0.001* | *10.2* | *86.28%* | *2.5426* | *5.7751* | *26.1114* | *196608* | *0.6%* | *1.019* |
| *MLP* | *27* | *1e-05* | *0.001* | *20.5* | *95.05%* | *2.5426* | *3.7081* | *26.1114* | *196608* | *1.73%* | *1.002* |
| MLP | 27 | 8e-06 | 0.001 | 21.3 | 93.55% | 2.5426 | 4.0623 | 26.1114 | 196608 | 0.66% | 0.988 |
| MLP | 27 | 6e-06 | 0.00075 | 26.4 | 91.19% | 2.5426 | 4.6185 | 26.1114 | 196608 | 0.57% | 0.973 |
| MLP | 27 | 5.5e-06 | 0.00075 | 18.1 | 85.53% | 2.5426 | 5.9522 | 26.1114 | 196608 | 0.58% | 0.994 |
| MLP | 27 | 3e-06 | 0.00075 | 26.9 | 90.82% | 2.5426 | 4.706 | 26.1114 | 196608 | 0.98% | 1.024 |
| *Attn* | *6* | *7e-05* | *0.00075* | *15.4* | *69.89%* | *2.5426* | *2.5989* | *2.7295* | *196608* | *96.78%* | *0.72* |
| *Attn* | *6* | *5e-05* | *0.00075* | *26.4* | *78.08%* | *2.5426* | *2.5836* | *2.7295* | *196608* | *98.97%* | *0.777* |
| *Attn* | *6* | *3e-05* | *0.00075* | *54.6* | *85.42%* | *2.5426* | *2.5698* | *2.7295* | *196608* | *99.7%* | *0.846* |
| *Attn* | *13* | *7e-05* | *0.00075* | *22.6* | *60.79%* | *2.5426* | *2.5481* | *2.5566* | *196608* | *93.47%* | *0.721* |
| *Attn* | *13* | *5e-05* | *0.00075* | *36.5* | *65.45%* | *2.5426* | *2.5474* | *2.5566* | *196608* | *97.59%* | *0.786* |
| *Attn* | *13* | *3e-05* | *0.00075* | *68.8* | *81.03%* | *2.5426* | *2.5452* | *2.5566* | *196608* | *99.19%* | *0.804* |
| *Attn* | *20* | *9e-05* | *0.00075* | *10.8* | *68.98%* | *2.5426* | *2.5519* | *2.5726* | *196608* | *79.34%* | *0.715* |
| *Attn* | *20* | *8e-05* | *0.00075* | *12.3* | *72.48%* | *2.5426* | *2.5508* | *2.5726* | *196608* | *83.58%* | *0.723* |
| *Attn* | *20* | *7e-05* | *0.00075* | *15.9* | *75.83%* | *2.5426* | *2.5498* | *2.5726* | *196608* | *87.54%* | *0.755* |
| *Attn* | *20* | *6e-05* | *0.00075* | *18.7* | *78.38%* | *2.5426* | *2.5491* | *2.5726* | *196608* | *89.49%* | *0.759* |
| *Attn* | *20* | *5e-05* | *0.00075* | *25.1* | *82.96%* | *2.5426* | *2.5477* | *2.5726* | *196608* | *92.36%* | *0.786* |
| *Attn* | *20* | *4e-05* | *0.00075* | *32.6* | *85.95%* | *2.5426* | *2.5468* | *2.5726* | *196608* | *95.14%* | *0.802* |
| *Attn* | *20* | *3e-05* | *0.00075* | *50.3* | *89.52%* | *2.5426* | *2.5457* | *2.5726* | *196608* | *96.52%* | *0.841* |
| *Attn* | *20* | *2e-05* | *0.00075* | *97.3* | *92.52%* | *2.5426* | *2.5448* | *2.5726* | *196608* | *95.74%* | *0.878* |
| *Attn* | *20* | *1.5e-05* | *0.00075* | *148.6* | *95.01%* | *2.5426* | *2.5441* | *2.5726* | *196608* | *92.55%* | *0.867* |
| *Attn* | *20* | *1e-05* | *0.00075* | *329.7* | *96.57%* | *2.5426* | *2.5436* | *2.5726* | *196608* | *78.75%* | *0.895* |
| *Attn* | *27* | *0.0008* | *0.00075* | *0.0* | *121.03%* | *2.5426* | *2.4291* | *3.0819* | *196608* | *5.34%* | *1.009* |
| *Attn* | *27* | *0.0006* | *0.00075* | *0.0* | *121.63%* | *2.5426* | *2.4259* | *3.0819* | *196608* | *4.7%* | *1.007* |
| *Attn* | *27* | *0.0001* | *0.00075* | *9.7* | *126.97%* | *2.5426* | *2.3971* | *3.0819* | *196608* | *35.94%* | *0.829* |

Table 5: Gemma-7B Baseline SAEs (1024 sequence length) continued from Table 4.

| Site | Layer | Sparsity λ | LR | L0 | % CE Recovered | Clean CE Loss | SAE CE Loss | 0 Abl. CE Loss | Width | % Alive Features | Shrinkage γ |
|---|---|---|---|---|---|---|---|---|---|---|---|
| *Resid* | *6* | *0.0012* | *0.0003* | *2.2* | *95.55%* | *2.5426* | *3.1483* | *16.1549* | *131072* | *93.94%* | *1.006* |
| *Resid* | *6* | *0.001* | *0.0003* | *3.0* | *96.67%* | *2.5426* | *2.9954* | *16.1549* | *131072* | *96.24%* | *1.006* |
| *Resid* | *6* | *0.0008* | *0.0003* | *4.3* | *97.83%* | *2.5426* | *2.8382* | *16.1549* | *131072* | *97.52%* | *1.003* |
| *Resid* | *6* | *0.0006* | *0.0003* | *7.0* | *98.76%* | *2.5426* | *2.7108* | *16.1549* | *131072* | *98.3%* | *0.996* |
| *Resid* | *6* | *0.0004* | *0.0003* | *14.3* | *99.35%* | *2.5426* | *2.6312* | *16.1549* | *131072* | *98.68%* | *0.996* |
| *Resid* | *6* | *0.0002* | *0.0003* | *45.9* | *99.77%* | *2.5426* | *2.5735* | *16.1549* | *131072* | *99.51%* | *0.999* |
| Resid | 6 | 2e-05 | 0.0003 | 95.2 | 98.62% | 2.5426 | 2.7302 | 16.1549 | 131072 | 45.13% | 1.148 |
| Resid | 6 | 4e-05 | 0.0003 | 144.0 | 99.35% | 2.5426 | 2.6313 | 16.1549 | 131072 | 36.05% | 1.038 |
| Resid | 6 | 8e-06 | 0.0003 | 177.5 | 99.29% | 2.5426 | 2.6386 | 16.1549 | 131072 | 53.36% | 1.086 |
| *Resid* | *6* | *0.0001* | *0.0003* | *131.8* | *99.94%* | *2.5426* | *2.5511* | *16.1549* | *131072* | *99.47%* | *1.005* |
| Resid | 6 | 8e-05 | 0.0003 | 153.2 | 99.93% | 2.5426 | 2.5524 | 16.1549 | 131072 | 98.14% | 0.984 |
| Resid | 6 | 6e-05 | 0.0003 | 215.7 | 99.93% | 2.5426 | 2.5521 | 16.1549 | 131072 | 93.91% | 0.982 |
| Resid | 6 | 4e-05 | 0.0003 | 284.5 | 99.62% | 2.5426 | 2.5948 | 16.1549 | 131072 | 84.71% | 2.56 |
| Resid | 6 | 2e-05 | 0.0003 | 801.3 | 99.82% | 2.5426 | 2.5673 | 16.1549 | 131072 | 91.71% | 1.272 |
| Resid | 6 | 8e-06 | 0.0003 | -288.2 | 99.7% | 2.5426 | 2.5835 | 16.1549 | 131072 | 85.02% | 1.006 |
| *Resid* | *13* | *0.0008* | *0.0003* | *5.4* | *98.3%* | *2.5426* | *3.0149* | *30.3588* | *131072* | *98.15%* | *1.008* |
| *Resid* | *13* | *0.0005* | *0.0003* | *13.1* | *99.25%* | *2.5426* | *2.7514* | *30.3588* | *131072* | *98.71%* | *0.998* |
| *Resid* | *13* | *0.0003* | *0.0003* | *31.8* | *99.62%* | *2.5426* | *2.6483* | *30.3588* | *131072* | *99.31%* | *0.992* |
| *Resid* | *13* | *0.0002* | *0.0003* | *62.6* | *99.76%* | *2.5426* | *2.6083* | *30.3588* | *131072* | *99.69%* | *0.993* |
| *Resid* | *13* | *0.0002* | *0.0003* | *63.7* | *99.77%* | *2.5426* | *2.6067* | *30.3588* | *131072* | *99.68%* | *0.997* |
| *Resid* | *13* | *0.0001* | *0.0003* | *146.1* | *99.87%* | *2.5426* | *2.5788* | *30.3588* | *131072* | *67.47%* | *1.056* |
| Resid | 13 | 0.0001 | 0.0003 | 96.8 | 99.64% | 2.5426 | 2.6421 | 30.3588 | 131072 | 64.18% | 0.934 |
| *Resid* | *20* | *0.001* | *0.0003* | *8.2* | *96.15%* | *2.5426* | *3.1995* | *19.5891* | *131072* | *96.49%* | *1.004* |
| *Resid* | *20* | *0.0009* | *0.0003* | *10.0* | *96.7%* | *2.5426* | *3.1059* | *19.5891* | *131072* | *96.89%* | *1.003* |
| *Resid* | *20* | *0.0008* | *0.0003* | *12.3* | *97.14%* | *2.5426* | *3.0293* | *19.5891* | *131072* | *97.46%* | *0.997* |
| *Resid* | *20* | *0.0007* | *0.0003* | *15.6* | *97.7%* | *2.5426* | *2.9353* | *19.5891* | *131072* | *98.02%* | *0.997* |
| *Resid* | *20* | *0.0005* | *0.0003* | *29.3* | *98.62%* | *2.5426* | *2.7775* | *19.5891* | *131072* | *98.66%* | *1.016* |
| Resid | 20 | 0.0005 | 0.0003 | 28.0 | 98.53% | 2.5426 | 2.7931 | 19.5891 | 131072 | 98.73% | 0.997 |
| Resid | 20 | 0.0005 | 0.0003 | 28.5 | 98.58% | 2.5426 | 2.7844 | 19.5891 | 131072 | 98.67% | 1.004 |
| *Resid* | *20* | *0.0003* | *0.0003* | *67.3* | *99.3%* | *2.5426* | *2.6611* | *19.5891* | *131072* | *99.33%* | *1.013* |
| *Resid* | *20* | *0.0002* | *0.0003* | *123.4* | *99.58%* | *2.5426* | *2.6139* | *19.5891* | *131072* | *99.69%* | *1.01* |
| *Resid* | *20* | *0.0001* | *0.0003* | *212.1* | *99.65%* | *2.5426* | *2.6024* | *19.5891* | *131072* | *55.01%* | *1.04* |
| *Resid* | *27* | *0.003* | *0.0003* | *17.3* | *81.66%* | *2.5426* | *4.3602* | *12.4534* | *131072* | *28.57%* | *1.001* |
| *Resid* | *27* | *0.002* | *0.0003* | *25.9* | *85.26%* | *2.5426* | *4.0033* | *12.4534* | *131072* | *31.98%* | *0.999* |
| *Resid* | *27* | *0.001* | *0.0003* | *54.4* | *90.26%* | *2.5426* | *3.5081* | *12.4534* | *131072* | *33.58%* | *1.008* |
| *MLP* | *6* | *0.0004* | *0.0003* | *4.0* | *73.71%* | *2.5426* | *2.604* | *2.7764* | *131072* | *98.69%* | *1.009* |
| *MLP* | *6* | *0.0001* | *0.0003* | *45.2* | *89.13%* | *2.5426* | *2.568* | *2.7764* | *131072* | *96.23%* | *0.998* |
| *MLP* | *6* | *7e-05* | *0.0003* | *106.0* | *90.67%* | *2.5426* | *2.5644* | *2.7764* | *131072* | *87.51%* | *1.0* |
| *MLP* | *13* | *9e-05* | *0.0003* | *36.0* | *76.36%* | *2.5426* | *2.5533* | *2.5878* | *131072* | *99.87%* | *1.002* |
| MLP | 13 | 9e-05 | 0.0003 | 36.1 | 76.25% | 2.5426 | 2.5533 | 2.5878 | 131072 | 99.91% | 1.004 |
| *MLP* | *13* | *8e-05* | *0.0003* | *48.9* | *78.71%* | *2.5426* | *2.5522* | *2.5878* | *131072* | *99.72%* | *1.007* |
| *MLP* | *13* | *7e-05* | *0.0003* | *69.7* | *82.15%* | *2.5426* | *2.5506* | *2.5878* | *131072* | *99.77%* | *1.01* |
| MLP | 13 | 7e-05 | 0.0003 | 67.0 | 81.24% | 2.5426 | 2.5511 | 2.5878 | 131072 | 99.61% | 0.997 |

Table 6: Gemma-7B Gated SAEs (1024 sequence length). Continued in Table 7.

| Site | Layer | Sparsity $\lambda$ | LR | L0 | % CE Recovered | Clean CE Loss | SAE CE Loss | 0 Abl. CE Loss | Width | % Alive Features | Shrinkage $\gamma$ |
|---|---|---|---|---|---|---|---|---|---|---|---|
| MLP | 13 | 5e-05 | 0.0003 | 196.4 | 85.54% | 2.5426 | 2.5491 | 2.5878 | 131072 | 76.56% | 1.003 |
| MLP | 13 | 3e-05 | 0.0003 | 766.5 | 93.04% | 2.5426 | 2.5457 | 2.5878 | 131072 | 86.81% | 1.033 |
| MLP | 20 | 0.00019 | 0.0003 | 24.4 | 87.81% | 2.5426 | 2.5603 | 2.6881 | 131072 | 99.91% | 1.004 |
| MLP | 20 | 0.00016 | 0.0003 | 32.7 | 89.16% | 2.5426 | 2.5583 | 2.6881 | 131072 | 99.94% | 1.004 |
| MLP | 20 | 0.00015 | 0.0003 | 36.4 | 89.63% | 2.5426 | 2.5577 | 2.6881 | 131072 | 99.95% | 1.002 |
| MLP | 20 | 0.00014 | 0.0003 | 40.8 | 89.73% | 2.5426 | 2.5575 | 2.6881 | 131072 | 99.96% | 1.0 |
| MLP | 20 | 0.00013 | 0.0003 | 46.6 | 90.3% | 2.5426 | 2.5567 | 2.6881 | 131072 | 99.95% | 1.002 |
| MLP | 20 | 0.00012 | 0.0003 | 53.5 | 90.99% | 2.5426 | 2.5557 | 2.6881 | 131072 | 99.99% | 1.001 |
| MLP | 20 | 0.0001 | 0.0003 | 74.9 | 91.42% | 2.5426 | 2.5551 | 2.6881 | 131072 | 99.99% | 0.999 |
| MLP | 20 | 9e-05 | 0.0003 | 91.2 | 92.01% | 2.5426 | 2.5542 | 2.6881 | 131072 | 99.9% | 0.998 |
| MLP | 20 | 8e-05 | 0.0003 | 111.3 | 93.3% | 2.5426 | 2.5523 | 2.6881 | 131072 | 100.0% | 1.0 |
| MLP | 20 | 1.1e-05 | 0.0003 | -91.1 | 103.85% | 2.5426 | 2.537 | 2.6881 | 131072 | 46.33% | 1.005 |
| MLP | 27 | 0.0012 | 0.0003 | 20.3 | 94.14% | 2.5426 | 3.9232 | 26.1114 | 131072 | 5.8% | 1.003 |
| MLP | 27 | 0.001 | 0.0003 | 23.1 | 96.01% | 2.5426 | 3.4834 | 26.1114 | 131072 | 6.13% | 0.995 |
| MLP | 27 | 0.0008 | 0.0003 | 27.3 | 96.47% | 2.5426 | 3.3747 | 26.1114 | 131072 | 5.18% | 1.005 |
| MLP | 27 | 0.0003 | 0.0003 | 59.3 | 99.07% | 2.5426 | 2.7627 | 26.1114 | 131072 | 3.89% | 1.002 |
| MLP | 27 | 0.0002 | 0.0003 | 80.9 | 98.19% | 2.5426 | 2.969 | 26.1114 | 131072 | 3.64% | 1.006 |
| MLP | 27 | 0.000175 | 0.0003 | 89.7 | 97.35% | 2.5426 | 3.1678 | 26.1114 | 131072 | 3.89% | 1.008 |
| MLP | 27 | 0.00015 | 0.0003 | 108.5 | 98.87% | 2.5426 | 2.8093 | 26.1114 | 131072 | 3.54% | 1.002 |
| MLP | 27 | 0.000135 | 0.0003 | 103.6 | 98.33% | 2.5426 | 2.9365 | 26.1114 | 131072 | 3.75% | 0.997 |
| Attn | 6 | 0.0007 | 0.0003 | 8.9 | 82.28% | 2.5426 | 2.5757 | 2.7295 | 131072 | 93.49% | 1.015 |
| Attn | 6 | 0.0005 | 0.0003 | 16.4 | 85.54% | 2.5426 | 2.5696 | 2.7295 | 131072 | 95.16% | 1.014 |
| Attn | 6 | 0.0003 | 0.0003 | 38.7 | 88.69% | 2.5426 | 2.5637 | 2.7295 | 131072 | 97.63% | 1.015 |
| Attn | 13 | 0.0012 | 0.0003 | 2.9 | 46.05% | 2.5426 | 2.5502 | 2.5566 | 131072 | 63.06% | 1.042 |
| Attn | 13 | 0.0006 | 0.0003 | 13.2 | 76.64% | 2.5426 | 2.5459 | 2.5566 | 131072 | 83.81% | 1.0 |
| Attn | 13 | 0.0004 | 0.0003 | 28.1 | 63.78% | 2.5426 | 2.5477 | 2.5566 | 131072 | 89.64% | 0.992 |
| Attn | 13 | 0.0002 | 0.0003 | 95.1 | 82.86% | 2.5426 | 2.545 | 2.5566 | 131072 | 97.05% | 0.993 |
| Attn | 13 | 4e-05 | 0.0003 | 1079.5 | 93.95% | 2.5426 | 2.5434 | 2.5566 | 131072 | 64.6% | 1.002 |
| Attn | 13 | 2e-05 | 0.0003 | -635.1 | 87.73% | 2.5426 | 2.5443 | 2.5566 | 131072 | 92.21% | 1.003 |
| Attn | 20 | 0.0012 | 0.0003 | 2.1 | 64.17% | 2.5426 | 2.5533 | 2.5726 | 131072 | 72.67% | 1.038 |
| Attn | 20 | 0.0006 | 0.0003 | 9.0 | 80.22% | 2.5426 | 2.5485 | 2.5726 | 131072 | 89.06% | 1.014 |
| Attn | 20 | 0.00055 | 0.0003 | 10.1 | 84.01% | 2.5426 | 2.5474 | 2.5726 | 131072 | 90.35% | 0.997 |
| Attn | 20 | 0.00045 | 0.0003 | 14.8 | 85.85% | 2.5426 | 2.5468 | 2.5726 | 131072 | 92.05% | 1.003 |
| Attn | 20 | 0.0004 | 0.0003 | 18.7 | 86.55% | 2.5426 | 2.5466 | 2.5726 | 131072 | 92.77% | 1.016 |
| Attn | 20 | 0.00035 | 0.0003 | 22.8 | 88.2% | 2.5426 | 2.5461 | 2.5726 | 131072 | 94.07% | 1.009 |
| Attn | 20 | 0.00025 | 0.0003 | 39.7 | 90.97% | 2.5426 | 2.5453 | 2.5726 | 131072 | 96.42% | 1.009 |
| Attn | 20 | 0.0002 | 0.0003 | 55.2 | 92.72% | 2.5426 | 2.5448 | 2.5726 | 131072 | 97.73% | 0.994 |
| Attn | 20 | 0.00015 | 0.0003 | 89.1 | 94.39% | 2.5426 | 2.5443 | 2.5726 | 131072 | 98.93% | 0.999 |
| Attn | 20 | 0.0001 | 0.0003 | 178.0 | 94.71% | 2.5426 | 2.5442 | 2.5726 | 131072 | 99.69% | 1.003 |
| Attn | 20 | 6e-05 | 0.0003 | 483.8 | 99.72% | 2.5426 | 2.5427 | 2.5726 | 131072 | 98.66% | 0.994 |
| Attn | 20 | 4e-05 | 0.0003 | 894.6 | 97.03% | 2.5426 | 2.5435 | 2.5726 | 131072 | 66.5% | 0.991 |
| Attn | 20 | 2e-05 | 0.0003 | -851.3 | 106.91% | 2.5426 | 2.5405 | 2.5726 | 131072 | 86.24% | 1.0 |
| Attn | 27 | 0.002 | 0.0003 | 6.6 | 100.37% | 2.5426 | 2.5406 | 3.0819 | 131072 | 56.82% | 1.008 |
| Attn | 27 | 0.001 | 0.0003 | 16.5 | 105.72% | 2.5426 | 2.5117 | 3.0819 | 131072 | 70.25% | 1.002 |
| Attn | 27 | 0.0007 | 0.0003 | 26.2 | 104.26% | 2.5426 | 2.5196 | 3.0819 | 131072 | 77.02% | 0.999 |

Table 7: Gemma-7B Gated SAEs (1024 sequence length). Continued from Table 6

| Site | Layer | Sparsity $\lambda$ | LR | L0 | % CE Recovered | Clean CE Loss | SAE CE Loss | 0 Abl. CE Loss | Width | % Alive Features | Shrinkage $\gamma$ |
|---|---|---|---|---|---|---|---|---|---|---|---|
| Attn | 4 | 8e-05 | 0.001 | 17.6 | 81.04% | 1.9699 | 1.9824 | 2.0361 | 196608 | 94.29% | 0.827 |
| Attn | 4 | 6e-05 | 0.001 | 24.2 | 84.12% | 1.9699 | 1.9804 | 2.0361 | 196608 | 95.76% | 0.848 |
| Attn | 4 | 3e-05 | 0.001 | 62.1 | 90.96% | 1.9699 | 1.9759 | 2.0361 | 196608 | 96.72% | 0.93 |
| Attn | 12 | 8e-05 | 0.001 | 16.1 | 51.88% | 1.9699 | 1.9907 | 2.0131 | 196608 | 65.73% | 0.78 |
| Attn | 12 | 6e-05 | 0.001 | 24.0 | 58.46% | 1.9699 | 1.9878 | 2.0131 | 196608 | 69.85% | 0.802 |
| Attn | 12 | 3e-05 | 0.001 | 75.0 | 72.84% | 1.9699 | 1.9816 | 2.0131 | 196608 | 73.04% | 0.848 |
| Attn | 16 | 0.00045 | 0.001 | 0.3 | -3.54% | 1.9699 | 2.0058 | 2.0046 | 49152 | 20.1% | 0.554 |
| Attn | 16 | 8e-05 | 0.001 | 14.6 | 67.69% | 1.9699 | 1.9811 | 2.0046 | 196608 | 64.35% | 0.798 |
| Attn | 16 | 3e-05 | 0.001 | 63.0 | 81.78% | 1.9699 | 1.9762 | 2.0046 | 196608 | 70.75% | 0.868 |
| Attn | 16 | 6e-05 | 0.001 | 20.8 | 72.07% | 1.9699 | 1.9796 | 2.0046 | 196608 | 69.92% | 0.813 |
| Attn | 16 | 0.0001 | 0.001 | 9.5 | 60.16% | 1.9699 | 1.9837 | 2.0046 | 49152 | 88.32% | 0.754 |
| Attn | 16 | 9e-05 | 0.001 | 11.3 | 62.62% | 1.9699 | 1.9829 | 2.0046 | 49152 | 89.87% | 0.769 |
| Attn | 20 | 6e-05 | 0.001 | 18.3 | 87.49% | 1.9698 | 1.9769 | 2.0269 | 196608 | 63.81% | 0.87 |
| Attn | 20 | 8e-05 | 0.001 | 13.6 | 85.63% | 1.9698 | 1.978 | 2.0269 | 196608 | 60.17% | 0.871 |
| Attn | 20 | 3e-05 | 0.001 | 52.0 | 91.92% | 1.9698 | 1.9744 | 2.0269 | 196608 | 65.83% | 0.899 |
| Attn | 28 | 3e-05 | 0.001 | 91.9 | 73.29% | 1.9698 | 1.9715 | 1.976 | 196608 | 71.36% | 0.817 |
| Attn | 28 | 6e-05 | 0.001 | 20.6 | 57.17% | 1.9698 | 1.9725 | 1.976 | 196608 | 64.79% | 0.771 |
| Attn | 28 | 8e-05 | 0.001 | 12.5 | 49.8% | 1.9698 | 1.9729 | 1.976 | 196608 | 55.92% | 0.747 |
| MLP | 4 | 3.5e-05 | 0.001 | 20.0 | 86.36% | 1.9698 | 1.9802 | 2.046 | 196608 | 95.6% | 0.954 |
| MLP | 4 | 1e-05 | 0.001 | 64.5 | 83.61% | 1.9698 | 1.9823 | 2.046 | 196608 | 42.92% | 0.977 |
| MLP | 4 | 2e-05 | 0.001 | 43.3 | 87.2% | 1.9698 | 1.9796 | 2.046 | 196608 | 74.78% | 0.986 |
| MLP | 12 | 3e-05 | 0.001 | 77.8 | 81.95% | 1.9698 | 1.9783 | 2.0167 | 196608 | 99.58% | 0.932 |

Table 8: Pythia-2.8B baseline SAEs (2048 sequence length). Continued in Table 9.

| Site | Layer | Sparsity $\lambda$ | LR | L0 | % CE Recovered | Clean CE Loss | SAE CE Loss | 0 Abl. CE Loss | Width | % Alive Features | Shrinkage $\gamma$ |
|---|---|---|---|---|---|---|---|---|---|---|---|
| MLP | 12 | 5e-05 | 0.001 | 28.2 | 76.01% | 1.9698 | 1.9811 | 2.0167 | 196608 | 99.45% | 0.909 |
| MLP | 12 | 7e-05 | 0.001 | 16.2 | 71.94% | 1.9698 | 1.983 | 2.0167 | 196608 | 99.14% | 0.883 |
| MLP | 16 | 2.5e-05 | 0.001 | 79.8 | 78.44% | 1.9698 | 1.9785 | 2.0098 | 196608 | 99.83% | 0.919 |
| MLP | 16 | 4e-05 | 0.001 | 29.0 | 72.83% | 1.9698 | 1.9807 | 2.0098 | 196608 | 99.82% | 0.923 |
| MLP | 16 | 3.5e-05 | 0.001 | 35.9 | 73.95% | 1.9698 | 1.9803 | 2.0098 | 196608 | 99.83% | 0.914 |
| MLP | 16 | 7.5e-05 | 0.001 | 11.2 | 65.88% | 1.9698 | 1.9835 | 2.0098 | 196608 | 99.45% | 0.884 |
| MLP | 16 | 4.5e-05 | 0.001 | 22.0 | 70.73% | 1.9698 | 1.9815 | 2.0098 | 196608 | 99.79% | 0.901 |
| MLP | 16 | 3e-05 | 0.001 | 54.9 | 76.5% | 1.9698 | 1.9792 | 2.0098 | 196608 | 99.86% | 0.947 |
| MLP | 20 | 3.5e-05 | 0.001 | 20.6 | 91.28% | 1.9698 | 1.9814 | 2.1022 | 196608 | 95.85% | 0.971 |
| MLP | 20 | 2.5e-05 | 0.001 | 25.4 | 91.64% | 1.9698 | 1.9809 | 2.1022 | 196608 | 90.15% | 0.964 |
| MLP | 20 | 7e-06 | 0.001 | 269.2 | 93.37% | 1.9698 | 1.9786 | 2.1022 | 196608 | 17.28% | 0.962 |
| MLP | 28 | 2.25e-05 | 0.001 | 95.2 | 79.05% | 1.9698 | 1.9792 | 2.0145 | 196608 | 99.81% | 0.941 |
| MLP | 28 | 4.5e-05 | 0.001 | 18.5 | 67.4% | 1.9698 | 1.9844 | 2.0145 | 196608 | 94.33% | 0.92 |
| MLP | 28 | 3e-05 | 0.001 | 37.0 | 71.12% | 1.9698 | 1.9827 | 2.0145 | 196608 | 92.72% | 0.932 |
| Resid | 4 | 3e-05 | 0.001 | 15.9 | 98.11% | 1.9699 | 2.1793 | 13.0434 | 49152 | 96.34% | 0.966 |
| Resid | 4 | 2e-05 | 0.001 | 28.1 | 98.67% | 1.9699 | 2.1174 | 13.0434 | 49152 | 97.0% | 0.974 |
| Resid | 4 | 1e-05 | 0.001 | 79.1 | 99.27% | 1.9699 | 2.0506 | 13.0434 | 49152 | 98.93% | 0.983 |
| Resid | 12 | 1e-05 | 0.001 | 128.7 | 97.68% | 1.9698 | 2.1712 | 10.6558 | 49152 | 52.7% | 0.951 |
| Resid | 12 | 3e-05 | 0.001 | 25.1 | 93.87% | 1.9698 | 2.5021 | 10.6558 | 49152 | 64.28% | 0.96 |
| Resid | 12 | 2e-05 | 0.001 | 52.1 | 96.34% | 1.9698 | 2.2874 | 10.6558 | 49152 | 67.39% | 0.979 |
| Resid | 16 | 2e-05 | 0.001 | 42.7 | 95.55% | 1.9698 | 2.4025 | 11.682 | 49152 | 68.44% | 0.975 |
| Resid | 16 | 1e-05 | 0.001 | 94.8 | 96.48% | 1.9698 | 2.3119 | 11.682 | 49152 | 36.81% | 0.94 |
| Resid | 16 | 1.5e-05 | 0.001 | 55.5 | 95.97% | 1.9698 | 2.3609 | 11.682 | 49152 | 59.52% | 0.95 |
| Resid | 16 | 3e-05 | 0.001 | 17.1 | 90.16% | 1.9698 | 2.9252 | 11.682 | 196608 | 9.91% | 0.932 |
| Resid | 16 | 5e-05 | 0.001 | 10.9 | 86.0% | 1.9698 | 3.3293 | 11.682 | 196608 | 8.82% | 0.929 |
| Resid | 16 | 8e-06 | 0.001 | 49.1 | 84.1% | 1.9698 | 3.5145 | 11.682 | 196608 | 1.06% | 0.946 |
| Resid | 20 | 7e-06 | 0.001 | 103.4 | 91.94% | 1.9698 | 2.6543 | 10.4578 | 49152 | 15.4% | 1.016 |
| Resid | 20 | 2e-05 | 0.001 | 33.4 | 90.97% | 1.9698 | 2.7363 | 10.4578 | 49152 | 46.57% | 0.986 |
| Resid | 20 | 4e-05 | 0.001 | 13.6 | 86.19% | 1.9698 | 3.1421 | 10.4578 | 49152 | 59.96% | 0.954 |
| Resid | 28 | 2e-05 | 0.001 | 21.0 | 95.09% | 1.9698 | 3.242 | 27.8663 | 49152 | 20.22% | 0.916 |
| Resid | 28 | 7e-06 | 0.001 | 109.2 | 97.45% | 1.9698 | 2.6298 | 27.8663 | 49152 | 20.65% | 1.021 |
| Resid | 28 | 1e-05 | 0.001 | 42.9 | 96.27% | 1.9698 | 2.9349 | 27.8663 | 49152 | 22.59% | 0.932 |

Table 9: Pythia-2.8B baseline SAEs (2048 sequence length). Continued from Table 8.

| Site | Layer | Sparsity $\lambda$ | LR | L0 | % CE Recovered | Clean CE Loss | SAE CE Loss | 0 Abl. CE Loss | Width | % Alive Features | Shrinkage $\gamma$ |
|------|-------|---------|------|------|----------------|---------------|-------------|----------------|-------|------------------|-----------|
| Attn | 4  | 0.0006 | 0.0003 | 38.2 | 92.85% | 1.9699 | 1.9746 | 2.0361 | 131072 | 93.76% | 1.006 |
| Attn | 4  | 0.0004 | 0.0003 | 69.8 | 94.82% | 1.9699 | 1.9733 | 2.0361 | 131072 | 96.29% | 1.0 |
| Attn | 4  | 0.0008 | 0.0003 | 24.7 | 90.94% | 1.9699 | 1.9759 | 2.0361 | 131072 | 91.45% | 1.007 |
| Attn | 12 | 0.0006 | 0.0003 | 64.5 | 82.04% | 1.9699 | 1.9776 | 2.0131 | 131072 | 74.48% | 0.99 |
| Attn | 12 | 0.001  | 0.0003 | 27.1 | 73.09% | 1.9699 | 1.9815 | 2.0131 | 131072 | 63.68% | 0.987 |
| Attn | 12 | 0.0008 | 0.0003 | 40.5 | 77.52% | 1.9699 | 1.9796 | 2.0131 | 131072 | 67.74% | 0.998 |
| Attn | 16 | 0.001  | 0.0003 | 17.2 | 79.67% | 1.9699 | 1.9769 | 2.0046 | 32768  | 89.76% | 0.988 |
| Attn | 16 | 0.0006 | 0.0003 | 39.1 | 87.21% | 1.9699 | 1.9743 | 2.0046 | 131072 | 80.93% | 0.985 |
| Attn | 16 | 0.0009 | 0.0003 | 20.8 | 81.8%  | 1.9699 | 1.9762 | 2.0046 | 32768  | 91.0%  | 0.993 |
| Attn | 16 | 0.0004 | 0.0003 | 77.2 | 90.56% | 1.9699 | 1.9732 | 2.0046 | 131072 | 85.48% | 0.987 |
| Attn | 16 | 0.0008 | 0.0003 | 25.0 | 83.57% | 1.9699 | 1.9756 | 2.0046 | 131072 | 79.41% | 0.993 |
| Attn | 16 | 0.0005 | 0.0003 | 57.8 | 88.63% | 1.9699 | 1.9738 | 2.0046 | 32768  | 96.08% | 0.992 |
| Attn | 20 | 0.0004 | 0.0003 | 71.2 | 96.25% | 1.9698 | 1.972  | 2.0269 | 131072 | 88.74% | 0.992 |
| Attn | 20 | 0.0006 | 0.0003 | 36.5 | 94.34% | 1.9698 | 1.973  | 2.0269 | 131072 | 85.88% | 0.986 |
| Attn | 20 | 0.0008 | 0.0003 | 24.0 | 93.05% | 1.9698 | 1.9738 | 2.0269 | 131072 | 83.05% | 0.994 |
| Attn | 28 | 0.0008 | 0.0003 | 27.8 | 73.39% | 1.9698 | 1.9715 | 1.976  | 131072 | 68.41% | 0.988 |
| Attn | 28 | 0.001  | 0.0003 | 17.7 | 68.35% | 1.9698 | 1.9718 | 1.976  | 131072 | 68.14% | 0.991 |
| Attn | 28 | 0.0006 | 0.0003 | 51.2 | 78.11% | 1.9698 | 1.9712 | 1.976  | 131072 | 72.44% | 0.986 |
| MLP  | 4  | 0.0006 | 0.0003 | 28.6 | 89.28% | 1.9698 | 1.978  | 2.046  | 131072 | 99.16% | 1.011 |
| MLP  | 4  | 0.0004 | 0.0003 | 66.5 | 92.74% | 1.9698 | 1.9754 | 2.046  | 131072 | 99.52% | 1.002 |
| MLP  | 4  | 0.0008 | 0.0003 | 15.8 | 87.13% | 1.9698 | 1.9796 | 2.046  | 131072 | 98.46% | 1.007 |
| MLP  | 12 | 0.001  | 0.0003 | 35.0 | 81.33% | 1.9698 | 1.9786 | 2.0167 | 131072 | 97.55% | 1.011 |
| MLP  | 12 | 0.002  | 0.0003 | 8.2  | 72.1%  | 1.9698 | 1.9829 | 2.0167 | 131072 | 94.68% | 1.002 |
| MLP  | 12 | 0.0008 | 0.0003 | 55.7 | 84.15% | 1.9698 | 1.9773 | 2.0167 | 131072 | 98.23% | 1.004 |

Table 10: Pythia-2.8B Gated SAEs (2048 sequence length). Continued in Table 11.

| Site | Layer | Sparsity λ | LR | L0 | % CE Recovered | Clean CE Loss | SAE CE Loss | 0 Abl. CE Loss | Width | % Alive Features | Shrinkage γ |
|------|-------|-----------|-----|-----|----------------|---------------|-------------|----------------|-------|------------------|-------------|
| MLP | 16 | 0.0008 | 0.0003 | 51.0 | 80.32% | 1.9698 | 1.9777 | 2.0098 | 131072 | 99.05% | 1.002 |
| MLP | 16 | 0.0016 | 0.0003 | 12.4 | 70.76% | 1.9698 | 1.9815 | 2.0098 | 131072 | 97.38% | 1.005 |
| MLP | 16 | 0.0007 | 0.0003 | 70.1 | 82.09% | 1.9698 | 1.977 | 2.0098 | 131072 | 99.32% | 1.001 |
| MLP | 16 | 0.0014 | 0.0003 | 16.1 | 72.62% | 1.9698 | 1.9808 | 2.0098 | 131072 | 97.48% | 1.007 |
| MLP | 16 | 0.0012 | 0.0003 | 21.9 | 75.12% | 1.9698 | 1.9798 | 2.0098 | 131072 | 98.18% | 1.012 |
| MLP | 16 | 0.0009 | 0.0003 | 38.3 | 78.41% | 1.9698 | 1.9785 | 2.0098 | 131072 | 98.72% | 0.993 |
| MLP | 20 | 0.0008 | 0.0003 | 51.0 | 94.28% | 1.9698 | 1.9774 | 2.1022 | 131072 | 99.06% | 1.007 |
| MLP | 20 | 0.0012 | 0.0003 | 22.1 | 92.53% | 1.9698 | 1.9797 | 2.1022 | 131072 | 97.97% | 1.0 |
| MLP | 20 | 0.001 | 0.0003 | 30.9 | 93.27% | 1.9698 | 1.9788 | 2.1022 | 131072 | 98.39% | 1.003 |
| MLP | 28 | 0.001 | 0.0003 | 47.7 | 79.96% | 1.9698 | 1.9788 | 2.0145 | 131072 | 98.76% | 1.004 |
| MLP | 28 | 0.0008 | 0.0003 | 82.1 | 83.68% | 1.9698 | 1.9771 | 2.0145 | 131072 | 98.48% | 1.002 |
| MLP | 28 | 0.0015 | 0.0003 | 21.3 | 73.3% | 1.9698 | 1.9818 | 2.0145 | 131072 | 97.58% | 1.004 |
| Resid | 4 | 0.0008 | 0.0003 | 70.7 | 99.5% | 1.9699 | 2.0257 | 13.0434 | 32768 | 99.68% | 0.996 |
| Resid | 4 | 0.001 | 0.0003 | 49.0 | 99.37% | 1.9699 | 2.0399 | 13.0434 | 32768 | 99.52% | 0.998 |
| Resid | 4 | 0.002 | 0.0003 | 16.2 | 98.83% | 1.9699 | 2.0998 | 13.0434 | 32768 | 98.72% | 1.001 |
| Resid | 12 | 0.004 | 0.0003 | 16.2 | 95.92% | 1.9698 | 2.3239 | 10.6558 | 32768 | 72.56% | 1.003 |
| Resid | 12 | 0.0016 | 0.0003 | 77.1 | 98.61% | 1.9698 | 2.0908 | 10.6558 | 32768 | 85.53% | 0.998 |
| Resid | 12 | 0.002 | 0.0003 | 52.8 | 98.2% | 1.9698 | 2.1261 | 10.6558 | 32768 | 83.41% | 1.0 |
| Resid | 16 | 0.003 | 0.0003 | 37.5 | 97.46% | 1.9698 | 2.2162 | 11.682 | 32768 | 78.14% | 1.0 |
| Resid | 16 | 0.006 | 0.0003 | 12.5 | 94.29% | 1.9698 | 2.5249 | 11.682 | 32768 | 62.59% | 0.998 |
| Resid | 16 | 0.002 | 0.0003 | 71.5 | 98.33% | 1.9698 | 2.1324 | 11.682 | 32768 | 82.89% | 0.998 |
| Resid | 16 | 0.0025 | 0.0003 | 46.2 | 98.04% | 1.9698 | 2.1597 | 11.682 | 131072 | 38.15% | 0.993 |
| Resid | 16 | 0.0045 | 0.0003 | 18.9 | 96.55% | 1.9698 | 2.3045 | 11.682 | 131072 | 38.92% | 0.996 |
| Resid | 16 | 0.0015 | 0.0003 | 95.6 | 98.62% | 1.9698 | 2.104 | 11.682 | 131072 | 29.77% | 0.991 |
| Resid | 20 | 0.0075 | 0.0003 | 15.4 | 91.68% | 1.9698 | 2.6763 | 10.4578 | 32768 | 59.39% | 0.994 |
| Resid | 20 | 0.004 | 0.0003 | 38.7 | 95.09% | 1.9698 | 2.3866 | 10.4578 | 32768 | 65.15% | 0.995 |
| Resid | 20 | 0.003 | 0.0003 | 58.4 | 96.05% | 1.9698 | 2.3053 | 10.4578 | 32768 | 68.08% | 0.994 |
| Resid | 28 | 0.0075 | 0.0003 | 25.0 | 96.54% | 1.9698 | 2.8646 | 27.8663 | 32768 | 29.97% | 0.993 |
| Resid | 28 | 0.005 | 0.0003 | 46.6 | 97.58% | 1.9698 | 2.5973 | 27.8663 | 32768 | 40.94% | 1.008 |
| Resid | 28 | 0.004 | 0.0003 | 61.2 | 97.9% | 1.9698 | 2.5136 | 27.8663 | 32768 | 35.93% | 1.005 |

Table 11: Pythia-2.8B Gated SAEs (2048 sequence length). Continued from Table 10.

