# OpenReview forum: "Improving Sparse Decomposition of Language Model Activations with Gated Sparse Autoencoders"
_NeurIPS.cc/2024/Conference — NeurIPS 2024 poster_

### Official Review · Reviewer_58d7 · 2024-07-09

**Soundness:** 3
**Presentation:** 3
**Contribution:** 3
**Rating:** 6
**Confidence:** 3

**Summary:**

This work proposed a Gated Sparse Autoencoder (Gated SAE) to mitigate the standard SAEs' biases, such as shrinkage, which systematically underestimate the feature activations from SAEs. The key difference between Gated SAE and SAE is that the Gated SAE separates affine transformations within the encoder in order to decide which dictionary elements to use in a reconstruction loss, and estimate the coefficients of active elements, although with the 50% more computing required to achieve. Comprehensive experiments are conducted to compare and verify how good the Gated SAE is to standard SAE, including a blinded human study to rate and compare the interpretability of randomly sampled Gated and baseline SAE features.

**Strengths:**

- A new architecture of SAE inspired by GRU is proposed to include a gate mechanism to mitigate shrinkage bias
- Comprehensive quantitative experiments including ablation studies to evaluate the proposed Gated SAE compared to SAEs
- A human evaluation to rate randomly sampled features from Gated SAE and SAE

**Weaknesses:**

- It is not very straightforward to understand how well the features from Gated SAE are compared to SAE based on Figure 4. Some case studies based on the open-source SAE visualizer library [1] are required to help better understand this.
- It will be better to see more case studies on downstream tasks to compare Gated SAE and SAE, e.g., automatic circuit detection [2]

[1] C. McDougall. SAE Visualizer, 2024. https://github.com/callummcdougall/sae_vis

[2] Huben, Robert, et al. "Sparse Autoencoders Find Highly Interpretable Features in Language Models." The Twelfth International Conference on Learning Representations. 2023.

**Questions:**

- As mentioned in the weakness above, the interpretability analysis was conducted via human rating for randomly sampled feature visualizations. However, those visualizations are not included in the main body or appendix. It will be better to help understand how well the sampled features are when comparing Gated SAE and SAE.
- Although the Gated SAE has good performance on the loss recovered (fidelity) and relative reconstruction bias, it is still not clear whether features from Gated SAE are better than SAE under downstream tasks. It will make this work very solid if some analysis of mini downstream applications can be conducted, e.g., IOI task, greater-than task, etc.

---

> ### Author Rebuttal · Authors · 2024-08-06
>
> We are grateful for your thoughtful review and helpful feedback, and are pleased you found our experiments comparing Gated and baseline SAEs comprehensive.
>
> We indeed used sae-vis, which you referenced in your review, to produce the visualization used in the interpretability study, citing this library section 4.2 where we explain the interpretability methodology. We agree however that it would be helpful to share screenshots of these dashboards, and will add a new appendix to the paper including some examples (from both Gated and baseline SAEs). We do note though that – as shown in Figure 4 – the differences in interpretability between Gated and baseline SAEs are slight (if any), and it is unlikely that these dashboards will give much insight into systematic differences in the features found by the respective architectures. Nevertheless, by including these dashboards we hope to better bring to life the information raters had to go on when rating the interpretability of the features.
>
> Regarding analyzing the performance of Gated SAEs on downstream uses, we agree this would be excellent evidence for the utility of Gated SAEs (or SAEs in general). However, as we note in our Limitations & Future Work section, the science of evaluating SAEs in a manner that correlates well with downstream uses (which are still uncertain) and yet is scalable and objective is in its early stages. To take the paper referenced in the review (Huben at al., 2023) as an example, while it makes a valuable contribution to furthering the study of SAE evaluations, the metrics introduced there are either better for resolving differences between SAEs as a whole vs other decompositions like PCA (i.e. Figure 4, where the difference between SAEs and e.g. PCA is stark, but the differences between SAEs are small), boil down to metrics that were indeed considered in our interpretability study (i.e. feature monosemanticity and direct effects on logits) or involve performing a bespoke study of one circuit with one SAE, which makes it difficult to scale and compare several SAEs objectively. To be clear, our point is not to undervalue the contribution of Huben at al., but to explain why we felt it would be unreasonable to try to combine in a single paper both the architectural innovation of Gated SAEs with simultaneously making a meaningful advance in the science of evaluating SAEs with different architectures. In this light, we do believe that our paper – by introducing a novel SAE architecture and showing that it either matches or exceeds baseline ReLU SAEs on standard SAE evaluation metrics – provides a valuable contribution even without these additional evaluations.

---

> > ### Comment · Reviewer_58d7 · 2024-08-09
> > **Reply by Reviewer 58d7**
> >
> > Thanks for this helpful rebuttal regarding the weaknesses and questions.
> >
> > Although proposed Gated SAEs outperform other SAE baselines under the standard SAE evaluation metrics, I am still wondering whether there are some existing works which try to explore and propose a reasonable metric to measure different SAEs in the mech interp areas. To my understanding, the purpose of utilising SAEs in the mech interp area is to better understand and explain different features (monosemanticity or polysemanticity) from original LLMs' attention heads, MLPs and residual streams. By comparing the better SAEs evaluation metrics of Gated SAEs and other baselines and the slight differences between them, it seems like there exists a mismatch between the objective SAEs metrics and subjective interpretability evaluation results. It might be unclear how valuable the new proposed Gated SAEs or other designed SAEs contribute to the interpretability of different features and semantic meanings of different LLMs' internal components.

---

> ### Author Response · Authors · 2024-08-12
>
> Thanks for considering our rebuttal and for your response. We believe the consensus in the field – as argued in [1], an example of work that explores and proposes metrics for both interpretability and reconstruction fidelity – is that fidelity, sparsity and interpretability are all important criteria to be satisfied by a good decomposition. From this perspective, we think there is no mismatch: fidelity and interpretability are separate desiderata, and it is a valuable contribution to establish – as we have done in this paper – that a change of architecture improves one without damaging the other. Furthermore, we have established our key result – that Gated SAEs improve fidelity (at fixed sparsity) and maintain interpretability – using similar standards of evidence to those used in influential prior work such as [1]. For these reasons, we believe our paper stands on its own even without further, downstream task-oriented evaluations, although we nonetheless agree that such evaluations would be interesting and informative.
>
> [1] Bricken, Trenton, et al. “Towards Monosemanticity: Decomposing Language Models With Dictionary Learning.” Transformer Circuits Thread, 2023. https://transformer-circuits.pub/2023/monosemantic-features

---

> > ### Comment · Reviewer_58d7 · 2024-08-13
> > **Reply by Reviewer**
> >
> > Thanks for this explanation. I will raise my score.

---

### Official Review · Reviewer_Qtwo · 2024-07-10

**Soundness:** 3
**Presentation:** 4
**Contribution:** 3
**Rating:** 7
**Confidence:** 5

**Summary:**

The paper attempts to resolve the issue of feature shrinkage in sparse autoencoders (SAEs) by replacing the SAE ReLU activation function with a gated ReLU unit. The weight-tying scheme they use for the gated unit effectively turns it into a jump ReLU activation function.
They train gated SAEs and baseline SAEs on a one layer transformer, Pyhtia-2.8B and Gemma-7B. They find that gated SAEs eliminate systematic shrinkage, and consistently outperform baseline SAEs on the pareto-curve of sparsity, measured by the L0 pseudonorm, and faithfulness, measured by the model loss recovered relative to a zero ablation baseline.
They run various additional tests involving variations of the gated and baseline SAE architectures, including combinations of SAE dictionaries with the classic gradient pursuit algorithm for choosing sparse feature coefficients at inference time. They conclude that the Pareto improvement of their gated SAEs over their baseline SAEs is due in part to better feature dictionaries, in addition to better estimated feature coefficients.
They compare the subjective interpretability of 150 gated SAE and baseline SAE features in Pythia-2.8B and 192 features in Gemma-7B, using a blinded analysis of activating dataset examples. They find that the features were similarly interpretable.

**Strengths:**

The paper attempts to address a substantive practical problem with current SAE training methods.


The paper's proposed new architecture is evaluated extensively, and many detailed additional investigations on the individual effects of various parts of the gated SAE architecture are described in sections 5.1, 5.2 and Appendix D.


I find Appendix D interesting in its own right, since it shows quantitative comparisons between SAE methods and the classic gradient pursuit optimization algorithm, as well as mixing SAE feature dictionaries with gradient pursuit for sparse approximation of feature coefficients. I have not encountered such a comparison before.


For the most part, good documentation of all their process is provided, and the writing and presentation are very clear in general.

**Weaknesses:**

The paper does not really address the concern that gated SAEs may outperform baseline SAEs in part by implicitly widening the definition of what it means for ‘features’ to be represented in the model. As the paper itself notes in Appendix D, though other more powerful sparse coding algorithms greatly outperform SAEs in terms of reconstruction and sparsity, there are concerns that the greater expressivity of these techniques lets them find spurious ‘features’ that would not be accessible to the model’s own internal computations. An SAE can only find features that are represented in the sense that their current values can be read off with a single ReLU probe, while an inference time algorithm or a multi-layer probe may read off ‘feature’ values that the model itself could not possibly access using a single MLP layer. A gated ReLU is far less expressive than an algorithm like gradient pursuit, but more expressive than a ReLU. So to what extent do gated SAEs outperform baseline SAEs merely because they are implicitly working with a more relaxed definition of what it means for a feature to be represented in the model? Figure 6 in Appendix D incidentally investigates this somewhat, since it attempts to compare the quality of gated vs. baseline dictionaries independent of their coefficients. However, the results there seem inconsistent, with smaller performance gaps and baseline SAEs outperforming gated SAEs at higher L0. I think this issue of the representational power of the probe used is pretty central for contextualizing the results, and should at least have been discussed.

Throughout the paper, the authors present reconstruction scores for SAEs in terms of the fraction of model loss recovered compared to a zero-ablation baseline. I think this metric obscures vital information. Lowering CE loss from e.g. 4.5 to 4.0 is typically much easier than lowering it from 1.5 to 1.0. Thus, the same difference in loss recovered between two SAEs can correspond to very different gaps in SAE quality. Without the raw CE scores, there is no direct way to infer how large the gap is quantitatively. At minimum, these raw CE scores should be in the supplementals. Better yet, the recovered performance could additionally be reported in terms of the compute required to train a model with the same CE score, as suggested in https://arxiv.org/abs/2406.04093.

**Questions:**

Why are the raw CE loss recovered scores not in the paper? Since it is typically much harder to lower CE loss from e.g. 4.5 to 4.0 than from 1.5 to 1.0, it is difficult to evaluate the quality gap between baseline and gated SAEs, or the quality of the baseline SAEs, without these scores.

**Limitations:**

All addressed.

---

> ### Author Rebuttal · Authors · 2024-08-06
>
> Thank you for taking the time to review our paper, we are heartened that you agree that Gated SAEs attempt to address a substantive practical problem with current SAE training methods and that you find our evaluations and ablations extensive.
>
> Regarding your questions about the raw (unspliced model's) cross-entropy loss, we agree that this is useful information that we had omitted from the paper and thank you for drawing this to our attention. To remedy this, we will add a table in the appendix listing original ("raw") CE losses and the CE losses incurred when zero ablating MLP, attention and residual stream layers for the layers and models investigated in the paper, as shown in the PDF accompanying the main rebuttal, and we will reference this table from the captions for the fidelity-vs-sparsity figures presented in the main body of the paper.
>
> Turning to the other limitation you mention in the weaknesses section, we agree that it is possible for a more powerful dictionary learning method to attain better fidelity by expanding what it means for a feature to be represented in a model. However, we do not believe that Gated SAEs – particularly with the weight-tying scheme proposed in the paper – fall into this trap.
>
> Making explicit the connection between SAEs and probing that you alluded to, a single row of a vanilla (ReLU) SAE encoder corresponds to a pair of linear models: a linear classifier that determines when a feature is active and a linear regressor that determines the magnitude of an active feature. A single row of a Gated SAE encoder also effectively corresponds to the same pair of a linear classifier and a linear regressor, performing the same respective functions. The key difference between vanilla and Gated SAEs is that whereas vanilla SAEs insist that both the weights and biases of the classifier and regressor be exactly identical, the Gated SAE allows the biases to differ. (The weights are still the same, up to a rescaling, under the weight tying scheme introduced in the paper.)
>
> As explained in section 3.1, our contention is that even in the ideal case of a perfectly linearly represented feature, it is suboptimal to use the same bias in the classifier and regressor, i.e. to use the same bias determine whether a feature is active and measure its magnitude – because this requires unnecessarily trading off false positives in the classifier (which occurs when the bias is too low) against shrinkage in the regressor (which occurs when the bias is too high). From this perspective, Gated SAEs are removing an undesirable inductive bias in vanilla SAEs, while still using the simplest class of models (i.e. thresholded affine transformations) to detect and measure feature activations.  Perhaps to put it another way, any feature that is detectable and measurable by a Gated SAE should be similarly detectable and measurable by a vanilla SAE, albeit with a worse bias-variance trade-off.
>
> The previous paragraph notwithstanding, it is in part to assuage fears that Gated SAEs may be somehow "cheating" in order to obtain better fidelity that we performed the interpretability study described in section 4.2, which finds reassuringly that Gated SAEs are comparably interpretable to vanilla SAEs. Nowhere in our training algorithm is there any term that directly trains features to be human interpretable, and so the fact that they remain human interpretable anyway is a good sign that we have not over-optimized for reconstruction / sparsity at the cost of the thing we actually care about.
>
> Nevertheless, we admit in the limitations section that "while we have confirmed that Gated SAE features are comparably interpretable to baseline SAE features, it does not necessarily follow that Gated SAE decompositions are equally useful for mechanistic interpretability", i.e. that our evaluations do not conclusively show that Gated SAEs are comparably "useful" as vanilla SAEs in a practical sense, and agree that this is an important open question that needs resolving.

---

> > ### Comment · Reviewer_Qtwo · 2024-08-13
> > **Response**
> >
> > Thanks to the authors for their responses and updates.
> >
> > The raw CE scores have been provided, which addresses what I think was the biggest problem with evaluating the paper’s results. Converting the CE scores into the compute required to train a model with the same CE score under the assumption of conventional scaling curves holding would still be preferred. As a reader, I currently need to perform this conversion myself ad-hoc anyway to actually judge the results quantitatively.
> >
> > I don’t find the authors’ reply to my concern about the greater representational power of gated SAEs compared to vanilla SAEs wholly convincing. Whether we classify them as part of a ‘simplest class of models’ or not, a single gated ReLU can compute outputs a vanilla ReLU cannot. Meaning it is theoretically able to find ‘features’ a ReLU cannot.
> >
> > It may be the case that in practice, sparse ‘features’ tend to be represented in a way that makes this unconcerning. But that would be an assumption that would need to be concretely defined, and empirically tested. I still believe this deserves more discussion than it got.
> >
> > The features being just as interpretable as those of vanilla SAEs does not seem to me to address the concern of representational power either. Merely mapping the hidden activations back to the original token embedding of the prompt that produced them would also yield highly human-interpretable ‘features’. Yet such ‘features’ would not be useful for interpretability, since they don’t describe the structure of the hidden representation at the current layer of the model in a way that would be accessible to the model’s own computations. With no constraints on the definition of a ‘feature’, metrics like human interpretability of feature activations are not particularly meaningful.
> >
> > Ultimately the authors do conclude: ‘Nevertheless, we admit in the limitations section that "while we have confirmed that Gated SAE features are comparably interpretable to baseline SAE features, it does not necessarily follow that Gated SAE decompositions are equally useful for mechanistic interpretability", i.e. that our evaluations do not conclusively show that Gated SAEs are comparably "useful" as vanilla SAEs in a practical sense, and agree that this is an important open question that needs resolving.’, which I agree with.

---

> > > ### Author Response · Authors · 2024-08-13
> > >
> > > Thanks for considering our rebuttal and for your reply.
> > >
> > > We did consider translating CE losses into equivalent compute, but realized that it would be difficult to do this in a principled manner, because neither of the main model families used in the paper (Pythia and Gemma v1) were scaled compute optimally. We do nevertheless agree that – given a family of models scaled compute optimally – translating CE loss into FLOPs would provide a more intuitive meaning to the y-axes of our plots.
> > >
> > > Turning to the point about representational power, we indeed agree that Gated SAEs do have (slightly) more representational power than ReLU SAEs. However, our argument is that ReLU SAEs have insufficient representational power to faithfully recover representations that would be considered linear features in the first place, e.g. the linear representations (in the presence of interference) described in [1]. In the presence of interference between features, ReLU SAEs must choose between underestimating feature magnitudes or allowing more false detections, and as a result they can’t faithfully recover the original features even when they are linear. Gated SAEs on the other hand do not make this trade off, and thereby better capture the class of representations that are considered to be linear representations (under interference) in the first place.
> > >
> > > However, we acknowledge that this conceptual argument does not prove that Gated SAEs improve usefulness on downstream tasks in practice, and are glad you agree with the limitation we noted regarding Gated SAEs being comparably interpretable and more faithful not necessarily implying that they are equally (or more) useful. Given the excitement about SAEs and the research priorities of the mechanistic interpretability field, we anticipate future work that evaluates Gated SAEs on downstream tasks, shedding further light on these concerns.
> > >
> > > [1] Elhage, et al. “Toy Models of Superposition.” Transformer Circuits Thread, 2022. https://transformer-circuits.pub/2022/toy_model/index.html

---

### Official Review · Reviewer_1pM6 · 2024-07-13

**Soundness:** 4
**Presentation:** 3
**Contribution:** 3
**Rating:** 6
**Confidence:** 4

**Summary:**

This work introduces a new technique under mechanistic interpretability's sparse autoencoders. By using a less naive SAE, with a gating mechanism and a little extra computation, the paper shows a decent improvement over the baseline.

**Strengths:**

This work addresses the important issue of interpreting transformer-based LLMs and clearly demonstrates an interesting method. The mechanistic interpretability community will certainly find this work of interest.

The writing is well written and fairly easy to follow, the results are clearly presented, and all relevant aspects of the method are appropriately ablated

I liked the setup of the internal user study; I think future papers will follow the design of the study closely.

The work's cited throughout the manuscript are incredibly thorough.

**Weaknesses:**

While I generally like the paper, I have two primary concerns:

* The architecture and loss are somewhat difficult to understand. I did appreciate the pseudo-code in the appendix, but I feel for readers not familiar with SAEs may have a hard time, especially with the optimization-based design choices of weight tying. Perhaps explaining weight tying later in 3.2 would help. I would especially prefer if a few lines of pseudo code could be added in the main paper, next to figure two.
* The user study results. I don't mind the small change in means between the method and the baseline, but the explainable AI community has been around for a long time and the shift from studies with a few experts to larger cohorts has been the norm for a while now. Just because there's a rebranding to mechanistic interpretability doesn't mean this field should settle for underpowered studies. Nevertheless, I do find the study setup itself to be well articulated and a very useful starting point for future work in this area.

Minor:
Some of the design choices (weight-tying, no r_mag, etc) aren't well explained until the ablation where we find they are primarily for optimization. This could be motivated a little earlier, i.e. that the pareto improvement comes from the separation, and not those choices.

**Questions:**

None

**Limitations:**

Yes

---

> ### Author Rebuttal · Authors · 2024-08-06
>
> We are grateful for your review and valuable feedback, and are encouraged that you found the paper fairly easy to follow, with results clearly presented and key aspects of the method appropriately ablated.
>
> We appreciate that the explanation of the architecture and loss function is somewhat dense, and perhaps hard to follow for readers unfamiliar with SAEs. To some extent we are limited in how much we can expand these sections due to the paper page limit, but we would like to take your suggestions on board. Regarding your suggestion to move the description of the weight tying scheme later in 3.2, we propose to move this subsection (i.e. lines 114–127) after the paragraph on Training, and give it its own minor heading ("Tying weights to improve efficiency"). Regarding moving a few lines of pseudo-code next to Figure 2, it will be difficult to do this due to space constraints, but instead we will mention in the figure's caption a reference to the pseudo-code in the appendix, and are also trying to improve the figure's legibility in response to another review (which will hopefully help address your concern too, by making it less important to rely on the pseudo-code).
>
> Turning to the interpretability study, we believe in hindsight that – due to some of the wording in section 4.2 – there is a danger that the aims of the study could be misunderstood in a way that leads readers to conclude the study was underpowered. Our primary motivation for introducing the Gated architecture was to improve the trade-off between reconstruction fidelity and sparsity; in this context, the purpose of this interpretability study was to provide reassurance that this improvement in reconstruction fidelity does not come at the expense of a deterioration in interpretability. In other words, our aim was not necessarily to show that Gated SAEs are **more** interpretable than baseline SAEs but rather to show that they aren't less interpretable by a practically relevant margin. We found that even with this study size we were able to confidently exclude a meaningful deterioration in interpretability – the confidence interval for the mean difference was found to be [0, 0.26] – allowing us to confidently state that Gated SAEs are at least similarly interpretable to baseline SAEs. This is indeed the conclusion advertised in the abstract, introduction and conclusion sections of the paper. Nevertheless, we will change some of the sentences in section 4.2 to make the real aim of the study clearer – i.e. giving more prominence to the confidence interval on the mean difference in interpretability as the key result of the study – and less prominence to the question of whether Gated SAEs are more interpretable, for which we agree a bigger sample size would be needed to settle the question.

---

> > ### Comment · Reviewer_1pM6 · 2024-08-13
> >
> > Thank you authors, I am satisficed that this paper is of interest to the community. I will keep my score.

---

### Official Review · Reviewer_Hix2 · 2024-07-13

**Soundness:** 4
**Presentation:** 4
**Contribution:** 3
**Rating:** 7
**Confidence:** 3

**Summary:**

This paper introduces Gated Sparse Autoencoders (Gated SAEs), an improvement over standard sparse autoencoders (SAEs) for decomposing language model activations. The key idea is to separate the tasks of detecting which features are active and estimating their magnitudes, allowing the sparsity penalty to be applied only to feature detection. Through experiments on language models up to 7B parameters, the authors show that Gated SAEs achieve better reconstruction fidelity for a given level of sparsity compared to baseline SAEs, while resolving issues like shrinkage. A human evaluation study finds Gated SAE features to be comparably interpretable to baseline features.

**Strengths:**

- A well-motivated architectural modification to SAEs that addresses key limitations
- Comprehensive empirical evaluation across multiple model sizes and activation sites demonstrating clear improvements over baseline SAEs
- Careful ablation studies and analysis to understand the source of improvements
- Human evaluation study to assess interpretability of learned features
- Thorough discussion of limitations and future work directions

**Weaknesses:**

- The presentation could be improved in some areas, particularly in explaining some of the technical details and metrics
- Some of the figures are quite dense and could be made more readable
- The human evaluation study, while valuable, has a relatively small sample size

**Questions:**

- Do you have any insights on how Gated SAEs might scale to even larger language models? Are there any potential limitations as model size increases?
- Have you explored using Gated SAEs for any downstream mechanistic interpretability tasks beyond the basic reconstruction and interpretability metrics? For example, does the improved reconstruction enable better circuit analysis?
- The weight tying scheme seems important for computational efficiency. Have you explored any alternative tying schemes? Is there a theoretical justification for why this particular scheme works well?

**Limitations:**

The authors provide a good discussion of limitations in the conclusion section. They appropriately note that their experiments were limited to specific model families and that further work is needed to evaluate usefulness for downstream interpretability tasks. They also acknowledge the subjective nature of the human interpretability study.
Regarding potential negative societal impacts, the authors do discuss this briefly in Appendix A. They note that advances in LM interpretability could potentially be misused, but argue that the current work poses minimal short-term risks. While this discussion is somewhat brief, it does address the key points and seems appropriate for the nature of this work.

---

> ### Author Rebuttal · Authors · 2024-08-06
>
> Thank you for your detailed review of our paper and your feedback and questions. We are glad you appreciated our explanation of the motivation behind the gated SAE architectural modification and found our evaluations and ablation studies thorough.
>
> We agree that the exposition is fairly dense and high context; in large part this was due to the need to stick within the page limit. Regarding the presentation of the metrics, although these are only briefly described in the main body, they are defined at further length in Appendix B. To help bring these definitions to life in the appendix, we will add illustrative examples to each definition. E.g., in the case of loss recovered we hope to make this metric clearer by adding, "For example, a SAE that increases cross-entropy loss from 2.4 to 2.5 when zero-ablation increases the loss to 5.0 would have a loss recovered of 96%". We are also considering how to improve Figure 2, which we think could be hard for someone unacquainted with SAEs to decipher at a quick glance. We would value your feedback on these proposals.
>
> Regarding the human evaluation study, we selected our sample size trying to balance the resources consumed by the study against its aim, which was principally to establish that Gated SAEs' superior fidelity over baselines does not come at the expense of interpretability. In other words, we set out to establish that there is no meaningful deterioration in interpretability, rather than trying to show that Gated SAEs are more interpretable. Although the discussion of the study in most of the paper (particularly the abstract, introduction and conclusion) are consistent with this aim, we acknowledge there are a few sentences in section 4 that may give the impression that the study was under-powered (because we fail to show that Gated SAEs are statistically significantly more interpretable), when in fact the main objective of the study – to show that the mean difference in interpretability between Gated and baseline SAEs is not negative by any practically important margin – was achieved, since we found the confidence interval for the difference in interpretability to be [0, 0.26], i.e. excluding significant deterioration in interpretability (even if it doesn't establish an improvement). We will reword key sentences from section 4 that are in danger of giving this incorrect impression, in order to emphasize that it is a meaningful deterioration in interpretability that we set out to measure, and that the study confidently shows that such a deterioration has not occurred.
>
> Turning to your questions:
> - We don't anticipate any issues with scaling Gated SAEs to even larger language models except for the issue acknowledged in the paper that they do require 50% more FLOPs per training step than baseline SAEs, holding width constant. In fact, we didn't actually see a 50% increase in step times during our experiments, because step times were so fast that data loading became a significant bottleneck (at least comparable to computing the loss and its gradient). As we scale to wider SAEs however, we anticipate that we should see the difference in training step time between Gated and baseline SAEs increase to 50%. (Note that our experiments suggest that even if we held compute constant to avoid this 50% increase, Gated SAEs would still outperform baseline SAEs.)
> - We view the use of SAEs for downstream tasks as a major research area for SAEs overall (Gated or otherwise), and would love to see follow-up work on it, as we have noted in our Limitations & Future Work section. Since submitting the paper, we have done some such work ourselves, as have many others (including with Gated SAEs), though we are hesitant to provide references due to the anonymity rules.
> - The key part of the weight tying scheme – tying the two encoder weight matrices – is theoretically motivated on the grounds that similar directions are likely useful for detecting whether a given feature is active and for detecting its magnitude if it is active; we did not come up with alternative ways to tie the encoder weights that seemed theoretically plausible and hence did not investigate them. We did look at whether r_mag is important for weight tying (this is discussed in section 5.1) and found that it seemed slightly beneficial. We also note in section 3.2 and Appendix H that with this weight tying scheme, Gated SAEs become equivalent to vanilla SAEs with the ReLU replaced by a "JumpReLU" activation function. From this perspective, one could view Gated SAEs with this weight tying scheme as a way to train a vanilla SAE with a JumpReLU activation function, overcoming the difficulties posed by the jump discontinuity in this activation function.

---

### Author Rebuttal · Authors · 2024-08-06

We thank the area chair and all our reviewers for taking the time to read our paper and for their insightful comments and suggestions.

We are encouraged by all four reviewers recommending that our paper be accepted, with reviewer Qtwo recognizing that our paper “attempts to address a substantive practical problem with current SAE training methods”. We perceive general agreement among the reviewers that the Gated SAE architecture is a useful and well-motivated innovation, and that the evaluations and ablations presented in the paper are thorough and comprehensive. We believe Gated SAEs significantly improve the quality of one of the most important tools in mechanistic interpretability – sparse autoencoders – and that this paper therefore deserves acceptance at the conference.

Two common criticisms in the reviewer feedback were: (1) that the presentation is dense in places and (2) that the sample size for the interpretability study should have been higher. Regarding (1), we have tried to address the specific concerns raised by the reviewers whilst satisfying page limit constraints, however we would welcome further feedback on our proposals.

Regarding criticism (2), we have realized that some of the text in Section 4.2 may give the incorrect impression we set out to show that Gated SAEs are more interpretable than baseline SAEs, when in fact the purpose of the study was to show that Gated SAEs don't sacrifice interpretability to achieve better reconstruction fidelity. We will therefore re-word the offending sentences in Section 4.2, to emphasize our main result that the mean difference in interpretability scores between Gated and baseline SAEs has a confidence interval of [0, 0.26], clearly excluding a practically relevant deterioration in interpretability.

Although we were (pleasantly) surprised to find the lower end of the confidence interval to be close to zero, and it is plausible that a bigger study might find that Gated SAEs are statistically significantly *more* interpretable than baseline SAEs, we are doubtful that such a finding would have much practical relevance. The effect size is likely to be fairly low, and it is uncertain whether a small increase in subjective interpretability would translate into a practically relevant increase in the usefulness of SAEs for downstream tasks, with the latter being the (still hard-to-measure) outcome the field really cares about.

Rather than focusing on subjective interpretability, multiple reviewers note that it would be better to see whether Gated SAEs are more useful for downstream tasks. We agree! Unfortunately, this is an area of weakness for the field as a whole, and as such we view this as an avenue for future work which would likely be its own paper.

In the accompanying PDF we include a new figure illustrating the dashboards used for the interpretability study, addressing feedback from reviewer 58d7, and a new table listing raw and zero-ablation CE losses to help put our loss recovered results into context, addressing feedback from reviewer Qtwo. Note that the dashboard figure will be printed landscape in the final paper (as shown in the PDF), as this is the only way we could make the dashboard legible due to its wide aspect ratio. The table will be in portrait format in the final paper.

---

### Decision · Program_Chairs · 2024-09-25

**Decision:**

Accept (poster)

**Comment:**

There was consensus among all reviewers that this paper belongs at the conference. The work is well-motivated, tackles the important topic of LLM interpretability, and provides extensive experimentation and ablation. Reviewers expressed concerns about the denseness of the paper, which I anticipate the authors will be able to redemy to some extent with the extra page for the camera-ready version.